# SIU3R: Simultaneous Scene Understanding and 3D Reconstruction Beyond Feature Alignment

**Qi Xu[1,2*], Dongxu Wei[2,3*†], Lingzhe Zhao[2], Wenpu Li[2], Zhangchi Huang[2,4],**
**Shunping Ji[1†], Peidong Liu[2†],**
[1]Wuhan University    [2]Westlake University
[3]Westlake Institute for Advanced Study    [4]Zhejiang University

**Project Website**: `https://insomniaaac.github.io/siu3r/`

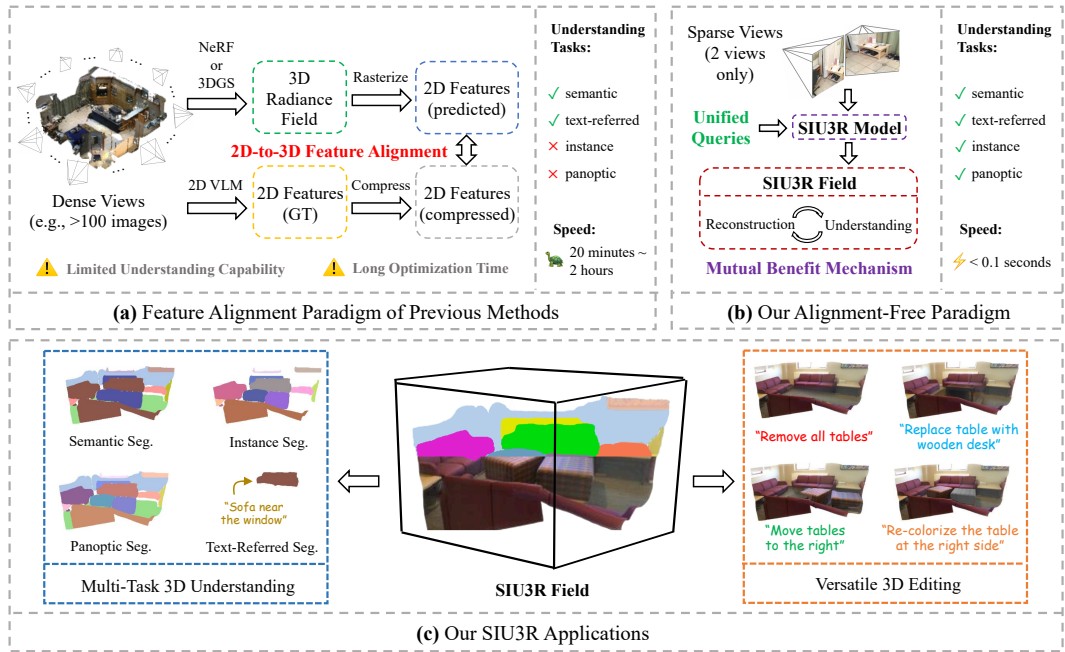

Figure 1: **Simultaneous Scene Understanding and 3D Reconstruction (SIU3R).** (a) 2D-to-3D Feature alignment paradigm of previous methods. (b) Alignment-free paradigm of our SIU3R method. (c) Versatile 3D reconstruction, understanding (multi-task 3D segmentation) and editing applications of our SIU3R method.

## Abstract

Simultaneous understanding and 3D reconstruction plays an important role in developing end-to-end embodied intelligent systems. To achieve this, recent approaches resort to 2D-to-3D feature alignment paradigm, which leads to limited 3D understanding capability and potential semantic information loss. In light of this, we propose SIU3R, the first alignment-free framework for generalizable simultaneous understanding and 3D reconstruction from unposed images. Specifically, SIU3R bridges reconstruction and understanding tasks via pixel-aligned 3D representation, and unifies multiple understanding (segmentation) tasks into a set of unified learnable queries, enabling native 3D understanding without the need of

---

[*]Qi Xu and Dongxu Wei contributed equally; [†] Corresponding author; This work was performed when Qi Xu was an intern at Westlake University.

39th Conference on Neural Information Processing Systems (NeurIPS 2025).

alignment with 2D models. To encourage collaboration between the two tasks with shared representation, we further conduct in-depth analyses of their mutual benefits, and propose two lightweight modules to facilitate their interaction. Extensive experiments demonstrate that our method achieves state-of-the-art performance not only on the individual tasks of 3D reconstruction and understanding, but also on the task of simultaneous understanding and 3D reconstruction, highlighting the advantages of our alignment-free framework and the effectiveness of the mutual benefit designs.

## 1    Introduction

In recent years, 3D scene reconstruction has achieved unprecedented quality and efficiency through differentiable rendering techniques[1, 2], while scene understanding has seen significant advancements via point cloud-based[3–5] or video-based[6–8] approaches powered by large-scale learning. Despite their individual successes, a critical gap remains: current frameworks often treat reconstruction and understanding as separate tasks, hindering the development of end-to-end embodied intelligence systems. In light of this, recent work [9–13] has endeavored to bridge these two tasks for simultaneous understanding and 3D reconstruction within a unified framework.

Current approaches to simultaneous understanding and 3D reconstruction typically follow a 2D-to-3D feature alignment paradigm (Fig.1 a). These methods[9–13] first extract 2D features from pre-trained 2D vision-language models (e.g., CLIP[14], LSeg[15]), then align and fuse them into 3D geometric representations (e.g., neural radiance fields[1] or 3D Gaussians[2]), through a per-scene iterative optimization. The resulting 3D language field jointly encodes scene geometry and semantics, enabling language-guided 3D segmentation via similarity matching with textual features. However, the per-scene optimization leads to heavy expenses for these methods, where each new scene requires dense image captures and hours of training to align 2D features with 3D structures through feature rasterization, severely hindering real-world deployment. The only exception is Large Spatial Model (LSM) [16], a very recent method that employs large reconstruction model and achieves generalizable 2D-to-3D feature alignment by predicting 3D Gaussians and their features in a single forward pass.

However, the aforementioned approaches inherently have the following limitations due to the nature of 2D-to-3D feature alignment. 1) Limited instance-level understanding: Existing feature alignment methods utilize off-the-shelf vision-language models that fall short in identifying instances, resulting in limited capability for instance-level understanding tasks such as instance and panoptic segmentation. 2) Information loss in feature compression: To efficiently embed 2D features into 3D representations and save the memory cost during feature rasterization, existing methods usually need to compress features to lower dimensions (e.g., from 512-dim to 64-dim [16]). Such compression discards fine-grained semantics, degrading performance on tasks requiring precise 3D understanding.

To address the challenges outlined above, we propose **SIU3R**, a novel generalizable framework achieving SIMULTANEOUS UNDERSTANDING and 3D RECONSTRUCTION beyond feature alignment (Fig.1 b). At its core component, SIU3R introduces a Unified Query Decoder and Mutual Benefit Mechanism that bridge 3D reconstruction and scene understanding within a unified SIU3R field through pixel-aligned 3D representation, which enables native 3D understanding through explicit 2D-to-3D lifting rather than implicit 2D-to-3D feature alignment. Specifically, we employ pixel-aligned representation for both reconstruction and understanding, ensuring the subsequently predicted 3D Gaussians and 2D masks can be naturally correlated with each other to enable 3D-level understanding. To ensure mask consistency across different views and understanding tasks, we propose Unified Query Decoder with a single set of learnable queries that shared across different views as well as multiple understanding tasks including semantic, instance, panoptic and text-referred 3D segmentation (Fig.1 c). To further investigate the bidirectional interaction between reconstruction and understanding in the context of their shared representation, we introduce two lightweight modules to encourage mutual benefits between the two tasks, reaching the best of both worlds.

In summary, our main contributions are as follows:

- We propose SIU3R, the first alignment-free framework for generalizable simultaneous understanding and 3D reconstruction, which bridges reconstruction and understanding via pixel-aligned 2D-to-3D lifting, and unifies multiple 3D understanding tasks (i.e., semantic, instance, panoptic and text-referred segmentation) into a set of unified learnable queries. This framework enables native

3D understanding without the need of alignment with 2D models, thereby avoiding limitations on 3D understanding imposed by 2D models and their feature compression.

- To our knowledge, this is the first work that conducts in-depth exploration of inter-task mutual benefits in the realm of simultaneous understanding and 3D reconstruction. To encourage the bidirectional promotion between the two tasks, we incorporate two lightweight modules into our pipeline and achieve significant performance improvements in both tasks.

- Extensive experiments on ScanNet[17] demonstrate that our method achieves state-of-the-art performance not only on the individual tasks of 3D reconstruction and understanding, but also on the integrated task of simultaneous understanding and 3D reconstruction, highlighting the advantages of our alignment-free framework and mutual benefit designs.

## 2  Related Work

**3D Reconstruction.** Recent advancements utilizing techniques like Neural Radiance Fields[1, 18–20] and 3D Gaussian Splatting[2, 21–23] have made remarkable progress in both reconstruction quality and rasterization speed. However, they typically require dense image captures as input and time-consuming per-scene optimization. To achieve reconstruction from sparse observations, large reconstruction models have emerged to incorporate 3D representations such as neural radiance fields[24–26], 3D Gaussians[27–33] and 3D point cloud[34, 35] into neural networks, enabling generalizable reconstruction in a feed-forward manner. Among these methods, DUSt3R[34] pioneers pose-free feed-forward 3D reconstruction given only two unposed images, avoiding the obstacles of acquiring camera poses in real world. By combining DUSt3R with additional 3D Gaussian head, [36, 37] can even surpass previous pose-required methods[29, 30] in novel view synthesis. Inspired by this, our method also adopts a pose-free paradigm in 3D reconstruction to ensure generality.

**Scene Understanding.** Remarkable progress has been witnessed in both 2D and 3D understanding. For 2D understanding, LSeg[15] achieves open-vocabulary semantic segmentation through feature alignment with a vision-language model (e.g., CLIP[14]), which lacks instance-level understanding capabilities such as instance or panoptic segmentation. In parallel, MaskFormer[38] treats semantic segmentation as the task of identifying region proposals of different classes, where a transformer model is used to derive regions from a set of learned class queries. As extensions to MaskFormer, subsequent works further achieve instance-level understanding [39, 40], as well as segmentation on videos[6–8]. Unfortunately, the above 2D methods can only understand scenes from specific camera perspectives, lacking a spatially consistent understanding at the 3D level. For 3D understanding, researchers[3–5, 41–43] typically take 3D point cloud clustered into super points as input and employ proposal-based method, similar to MaskFormer, to segment super points into 3D semantic regions or object instances. However, all of these 3D understanding methods rely on pre-scanned 3D point cloud and pre-processed super points, incurring significant costs in practical applications.

**Simultaneous Understanding and 3D Reconstruction.** Despite the individual successes of the above methods, they all treat reconstruction and understanding as separate tasks, limiting the potential of building end-to-end embodied intelligence systems. Therefore, recent advances propose to embed language features into neural radiance fields[9, 10, 44–46] or 3D Gaussians[11–13, 47–51] to empower the reconstructed 3D scenes with understanding capabilities. Given dense image captures of a scene, they typically process images into full-image[9–11, 13, 44–46] or object-wise[12, 47–51] features using 2D foundation models (e.g., CLIP[14], LSeg[15]), and then align 3D feature fields with 2D features by imposing losses on the feature rasterization process. Since these methods require time-consuming training for each scene, their efficiency is limited in practical applications. A very recent method Large Spatial Model (LSM)[16] eliminates this drawback by performing feature alignment within a large reconstruction model, achieving generalizable 3D reconstruction and understanding for the first time. Nevertheless, all of these methods follow 2D-to-3D feature alignment paradigm, which leads to limited understanding capability and semantic information loss.

## 3  Methodology

### 3.1  Problem Formulation and Pipeline

SIU3R processes sparse unposed multi-view images with corresponding camera intrinsics $\{\boldsymbol{I}^v, \boldsymbol{K}^v\}_{v=1}^{V}$, where $V \geq 2$ in our setting and denotes the number of input context views, and

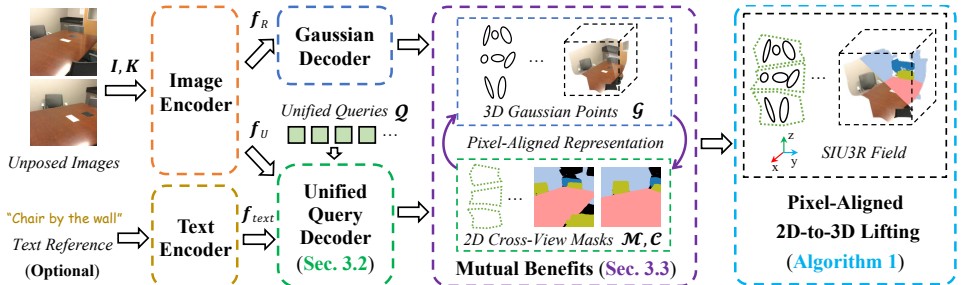

Figure 2: **Pipeline.** Our method consists of Image and Text Encoders for extracting multi-view and text features, Gaussian Decoder for decoding pixel-aligned 3D Gaussians, Unified Query Decoder for decoding pixel-aligned 2D cross-view masks, Mutual Benefit Mechanism for enabling bidirectional promotion between reconstruction and understanding tasks, Pixel-Aligned 2D-to-3D Lifting algorithm for obtaining SIU3R field that enables simultaneous understanding and 3D reconstruction.

learns a feed-forward network $\mathcal{F}_{\Theta,\mathcal{Q}}$ parameterized by network weights $\Theta$ and $N_q$ learnable unified queries $\mathcal{Q} = \{q_n\}_{n=1}^{N_q}$. The network establishes two key outputs: 1) pixel-aligned multi-view 3D Gaussians $\mathcal{G} = \{g_v^{ij}\}_{v,i,j=1}^{V,H,W}$ for 3D reconstruction, where $g = \{\mu, \alpha, r, s, c\}$ is a single gaussian primitive and $H, W$ specify the image resolution; 2) mask prediction logits $\mathcal{M} = \{m_n\}_{n=1}^{N_q}$ where $m_n \in \mathbb{R}^{V \times H \times W}$, and class prediction logits $\mathcal{C} = \{c_n\}_{n=1}^{N_q}$ where $c_n \in \mathbb{R}^{N_c}$ for scene understanding and $N_c$ indicates total classes including background. Formally, the learning objective implements the mapping:

$$\mathcal{F}_{\Theta,\mathcal{Q}} : \{I, K\} \mapsto \{\mathcal{G}, \mathcal{M}, \mathcal{C}\} \tag{1}$$

As illustrated in Fig.2, our pipeline comprises Image Encoder, Text Encoder, Unified Query Decoder (Sec.3.2), Gaussian Decoder, Mutual Benefit Mechanism (Sec.3.3) and Pixel Aligned 2D-to-3D Lifting Algorithm (Algo.1). We design our **Image Encoder** following [52]'s architecture as a Vision Transformer (ViT) enhanced with an adapter module. This configuration enables simultaneous extraction of geometry-focused features $\{f_R^v\}_{v=1}^V$ for reconstruction and semantic-focused features $\{f_U^v\}_{v=1}^V$ for understanding. The extracted semantic features $\{f_U^v\}_{v=1}^V$ along with object queries $\mathcal{Q}$ and optional text features $f_{text}$ extracted by CLIP text encoder[53] are fed into **Unified Query Decoder** to obtain pixel-aligned mask logits $\mathcal{M}$ and class logits $\mathcal{C}$ shared across all views and all understanding tasks, enabling cross-view and cross-task consistent mask predictions. At the same time, **Gaussian Decoder** consisting of a ViT decoder with a DPT head[54] takes geometry features $\{f_R^v\}_{v=1}^V$ as input and predicts pixel-aligned 3D Gaussians $\mathcal{G}$, which are aligned with the 2D cross-view masks. To fully exploit the inherent correlation regarding the shared representation between semantic predictions (i.e., $\mathcal{M}, \mathcal{C}$) and geometric predictions (i.e., $\mathcal{G}$), we further propose **Mutual Benefit Mechanism** to encourage collaboration between reconstruction and understanding. In particular, to promote understanding from reconstruction, we propose *Multi-View Mask Aggregation* module, which utilizes 3D geometric clues in $\mathcal{G}$ to aggregate semantic information from all views to improve cross-view consistency of $\mathcal{M}$ and $\mathcal{C}$. Moreover, to improve reconstruction by understanding, we introduce *Mask-Guided Geometry Refinement* module that leverages 2D masks to enforce intra-instance depth continuity for refining reconstructed 3D geometry. Finally, through **Pixel-Aligned 2D-to-3D Lifting**, we can obtain SIU3R field that supports simultaneous understanding and 3D reconstruction.

## 3.2 Unified Query Decoder

As shown in Fig.3 (a), we employ a set of learnable unified queries $\mathcal{Q}$ to jointly decode cross-view consistent masks for both instance and semantic segmentation tasks, where each query $q_n \in \mathcal{Q}$ explicitly denotes a potential object instance or semantic region that may appear in multiple views. To aggregate semantic logits from multi-view image features, we perform cross-attention between unified queries $\mathcal{Q}$ (query) and semantic-focused multi-view features $\{f_U\}_{v=1}^V$ (key/value), followed by self-attention layer that enables inter-query correlations. This cross-/self-attention block is stacked $L_1$ times to progressively consolidate semantic information across views, ultimately decoding them into multi-view mask logits $\mathcal{M} = \{m_{n,v}^{ij}\}_{n,v,i,j=1}^{N_q,V,H,W}$ and class logits $\mathcal{C} = \{c_n\}_{n=1}^{N_q}$ through linear projection. Benefiting from the shared representation between $\mathcal{M}$ and reconstructed multi-view 3D Gaussians $\mathcal{G}$, we can correlate each $m_{n,v}^{ij}$ with its 3D Gaussian counterpart $g_v^{ij}$, and lift it to 3D

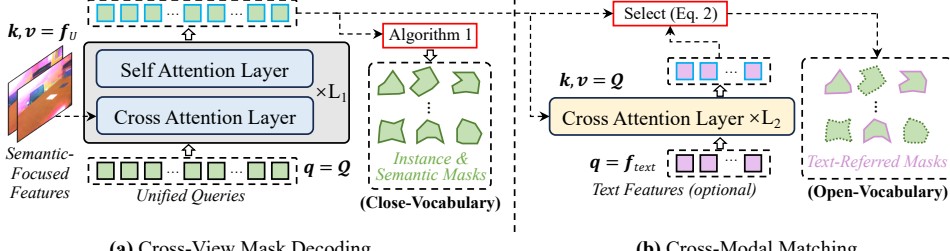

**(a)** Cross-View Mask Decoding      **(b)** Cross-Modal Matching

Figure 3: **Unified Query Decoder.** In (a), we employ multi-view semantic-focused features $\boldsymbol{f}_U$, unified queries $\mathcal{Q}$, and $L_1$ stacked cross-/self-attention layer blocks to decode cross-view instance and semantic masks. In (b), we employ $L_2$ stacked cross-attention layers to select the queries that best match the text features $\boldsymbol{f_{text}}$, and further derive them into text-referred masks.

space for multiple close-vocabulary 3D understanding tasks (i.e., instance, semantic and panoptic segmentation) according to Pixel-Aligned 2D-to-3D Lifting Algorithm derived in Algo.1.

As shown in Fig.3 (b), to enable text features $\boldsymbol{f}_{text} = \{\boldsymbol{f}_{text}^t\}_{t=1}^{N_t}$ as input for open-vocabulary 3D understanding, we further incorporate cross-modal matching module into Unified Query Decoder. Here we assume that our unified queries can perceive all potential object instances and semantic regions. What we need to do is then to identify the queries that best match the text features. Specifically, we employ $L_2$ cross-attention layers to enable interaction between $\boldsymbol{f}_{text}$ and $\mathcal{Q}$. Then for each text feature $\boldsymbol{f}_{text}^t$, we can match it with the most correlated query $\boldsymbol{q}_{text}^t$, and use its logit prediction to derive text-referred mask, which can also be lifted to 3D for open-vocabulary 3D understanding following Algo.1 same as close-vocabulary tasks. The matching process is derived as follows:

$$\boldsymbol{q}_{text}^t = \arg \max_{\boldsymbol{q}_n \in \mathcal{Q}} \left( \text{Attn}(\boldsymbol{f}_{text}^t, \mathcal{Q}) \cdot \boldsymbol{q}_n \right), \tag{2}$$

where $\text{Attn}(\cdot, \cdot)$ denotes cross-modal attention between text feature $\boldsymbol{f}_{text}^t$ (query) and object queries $\mathcal{Q}$ (key/value) $L_2$ times, followed by dot product operation with each query $\boldsymbol{q}_n \in \mathcal{Q}$.

We also introduce the following loss in training to enable matching supervision:

$$\mathcal{L}_{text} = \frac{1}{N_t} \sum_{t=1}^{N_t} \text{CrossEntropy}(\text{Softmax}(\text{Attn}(\boldsymbol{f}_{text}^t, \mathcal{Q}) \cdot \boldsymbol{q}_n), \boldsymbol{\delta}_n^{gt}), \tag{3}$$

where $\boldsymbol{\delta}_n^{gt}$ denotes ground-truth one-hot label that indicates the best matched query. We obtain $\boldsymbol{\delta}_n^{gt}$ by conducting Hungarian matching[39, 55] between $\mathcal{M}$ and ground-truth text-referred masks.

### 3.3 Mutual Benefit Mechanism

**Multi-View Mask Aggregation.** Due to potential large viewpoint changes, occlusions, or lighting differences between images from different camera views, the semantic information may vary significantly. Thus, the masks directly predicted by the Unified Query Decoder may exhibit inconsistencies across different views. A typical case is shown in Fig.4 (a), the Unified Query Decoder only considers cross-view semantic correlation of 2D pixels, without realizing that these pixels from different views are probably neighbors in 3D space for the same instance. As a result, the same instance is predicted

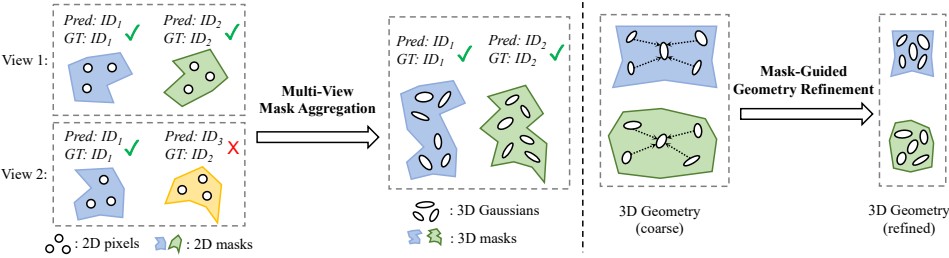

**(a)** $\mathbf{R} \rightarrow \mathbf{U}$: Reconstruction Helps Understanding      **(b)** $\mathbf{U} \rightarrow \mathbf{R}$: Understanding Helps Reconstruction

Figure 4: **Mutual Benefit Mechanism.** In (a), our Multi-View Mask Aggregation module utilizes reconstructed 3D Gaussians as geometry clues to improve cross-view mask consistency in 3D understanding. In (b), our Mask-Guided Geometry Refinement module employs segmentation masks as semantic clues to refine geometry of reconstructed 3D Gaussians.

**Algorithm 1** *Pixel-aligned 2D-to-3D lifting for simultaneous understanding and 3D recontruction.*

/* Model forward pass */
$\mathcal{G} \leftarrow$ Gaussian Decoder      ▷ Pixel-aligned 3D Gaussians
$\mathcal{Q}, \mathcal{M}, \mathcal{C} \leftarrow$ Unified Query Decoder    ▷ Last-layer hidden states of unified queries, pixel-aligned multi-view mask logits and class logits
$\boldsymbol{f}_{text} \leftarrow$ CLIP Text Encoder      ▷ Obtain textual features of human instructions (**Optional**)
/* Obtain predictions of valid queries, and select the query that matches with the text reference */
$\boldsymbol{kept\_qs} \leftarrow \max(\text{softmax}(\mathcal{C})) > \tau_c \bigcap \arg\max(\text{softmax}(\mathcal{C})) \neq \varnothing$    ▷ Kept queries of high confidences and valid classes
$text\_id \leftarrow$ Select$(\mathcal{Q}, \boldsymbol{f}_{text})$    ▷ Select the query that best matches the text reference following Eq. 2 (**Optional**)
$\mathcal{Q}', \mathcal{M}', \mathcal{C}' \leftarrow \mathcal{Q}[\boldsymbol{kept\_qs}], \mathcal{M}[\boldsymbol{kept\_qs}], \mathcal{C}[\boldsymbol{kept\_qs}]; \ N_q' \leftarrow \text{len}(\mathcal{Q}')$    ▷ Filter out queries of no interest
$\mathcal{C}', \mathcal{M}' \leftarrow \text{softmax}(\mathcal{C}'), \text{sigmoid}(\mathcal{M}')$    ▷ Turn logits into class confidence scores and multi-view query probability maps
$\mathcal{Z} \leftarrow \mathcal{C}'[\text{None}, :, :, \text{None}, \text{None}] * \mathcal{M}'[:, :, \text{None}, :, :]$    ▷ Calculate class-wise query probability maps $\mathcal{Z} \in \mathbb{R}^{V \times N_q' \times N_c \times H \times W}$
/* Multi-view mask aggregation */
$\mathcal{G}^* \leftarrow \boldsymbol{g}^{*\,ij}_v; \ \boldsymbol{g}^* = \{\boldsymbol{\mu}, \boldsymbol{\alpha}, \boldsymbol{r}, \boldsymbol{s}, \boldsymbol{z}\}$    ▷ Replace original 3D Gaussian color attribute $\boldsymbol{c}$ with semantic attribute $\boldsymbol{z}$ by spatial-indexing $\mathcal{Z}$
$\mathcal{Z} \leftarrow \text{rasterize}(\mathcal{G}^*)$    ▷ Fuse multi-view semantics in 3D and propagate them back to the original views via rasterization
/* Derive cross-view 2D masks */
$\boldsymbol{M}_q, \mathcal{I}_q \leftarrow \max(\mathcal{Z}, \dim = 1)$    ▷ Derive class-wise masks and maximum-probability query indices, $\boldsymbol{M}_q, \mathcal{I}_q \in \mathbb{R}^{V \times N_c \times H \times W}$
$\_, \boldsymbol{M}_{sem} \leftarrow \max(\boldsymbol{M}_q, \dim = 1)$    ▷ Derive multi-view semantic masks by maximizing class confidences, $\boldsymbol{M}_{sem} \in \mathbb{R}^{V \times H \times W}$
$\_, \boldsymbol{M}_{ins} \leftarrow \mathcal{I}_q[\boldsymbol{M}_{sem}]$    ▷ Derive multi-view instance masks by taking $\boldsymbol{M}_{sem}$ as class indices, $\boldsymbol{M}_{ins} \in \mathbb{R}^{V \times H \times W}$
/* 2D-to-3D mask lifting based on 3D Gaussians for multi-task 3D understanding */
$\mathcal{G}_{sem} \leftarrow \{\mathcal{G}_{sem\_id} \mid sem\_id \in [1, N_c]\}; \ \mathcal{G}_{sem\_id} \leftarrow \{\boldsymbol{g}^{ij}_v \mid \boldsymbol{M}^{v, ij}_{sem} = sem\_id\}$    ▷ 3D Gaussian-based semantic segmentation
$\mathcal{G}_{ins} \leftarrow \{\mathcal{G}_{ins\_id} \mid ins\_id \in [1, N_q]\}; \ \mathcal{G}_{ins\_id} \leftarrow \{\boldsymbol{g}^{ij}_v \mid \boldsymbol{M}^{v, ij}_{ins} = ins\_id\}$    ▷ 3D Gaussian-based instance segmentation
$\mathcal{G}_{pano} \leftarrow \{\mathcal{G}_{sem}, \mathcal{G}_{ins}\}$    ▷ 3D Gaussian-based panoptic segmentation
$\mathcal{G}_{text} \leftarrow \{\mathcal{G}_{ins\_id} \mid ins\_id = text\_id\}$    ▷ 3D Gaussian-based text-referred segmentation (**Optional**)

with different IDs across different views when semantic information alone is insufficient. Benefiting from the pixel-aligned representation shared between our predicted 2D masks and 3D Gaussians, we can use 3D Gaussians as additional geometric clues to fuse multi-view 2D semantic information in 3D space and propagate it back to the original views to avoid inconsistency. We call this "*Reconstruction Helps Understanding (R→U)*".

Specifically, we propose Multi-View Mask Aggregation module, which first lifts 2D semantic information (i.e., query logits $\mathcal{M}$ and $\mathcal{C}$) from different views to the 3D Gaussians $\mathcal{G}$ for fusion, and then propagate them back to these views through alpha-blending mechanism inherent in the 3D Gaussian rasterization. As highlighted in Algo.1, this module only introduces minor extra computation to the vanilla 2D-to-3D lifting algorithm during the inference phase, without the need for additional training.

Notably, another design choice for the above multi-view aggregation is to apply the 3D Gaussian rasterization earlier to $\boldsymbol{f}_U$ at the stage of feature encoding rather than the mask decoding, which is the core design of previous feature alignment-based methods[11, 12, 16]. However, due to extremely high dimension of 2D features, we have to perform feature compression before rasterization to avoid memory exhaustion, which inevitably corrupts the original semantic information. As demonstrated in Sec.4.3, such early aggregation not only requires additional training but also incurs significant memory overhead, yet its performance is far inferior to our training-free Multi-View Mask Aggregation.

**Mask-Guided Geometry Refinement.** In general, adjacent 2D pixels within the same object instance or semantic region should correspond to continuous positions in 3D space. Leveraging this prior knowledge, we can use our mask predictions as semantic clues to refine the reconstructed 3D geometries. We call this "*Understanding Helps Reconstruction (U→R)*". As shown in Fig.4 (b), the 3D Gaussians corresponding to adjacent pixels within the same instance may be far apart without refinement, which can lead to unsatisfactory coarse geometry. Thus, to make 3D Gaussians within the same mask to be more clustered, we propose Mask-Guided Geometry Refinement module, which utilizes masks as guidance to enforce intra-instance depth continuity based on the following loss:

$$\mathcal{L}_{cont} = \sum_{k=1}^{K} \sum_{p \in \boldsymbol{M}_k} ||D(p) - \frac{1}{|\mathcal{N}_p|} \sum_{q \in \mathcal{N}_p} D(q)||_2^2, \tag{4}$$

where $\boldsymbol{M}_k$ denotes pixels belonging to the $k$-th mask, $D(p)$ and $D(q)$ represent the depths at pixel $p$ and one of its neighbored pixel $q$, and $\mathcal{N}_p$ indicates the neighborhood of $p$ within the same instance.

### 3.4 Training Objective

Through holistic integration of components, our framework enables end-to-end optimization across the complete learning pipeline. The overall training objective is derived as follows:

$$\mathcal{L} = \lambda_1 ||\boldsymbol{I}(\mathcal{G}) - \hat{\boldsymbol{I}}|| + \lambda_2 \text{LPIPS}(\boldsymbol{I}(\mathcal{G}), \hat{\boldsymbol{I}}) + \lambda_3 \mathcal{L}_{mask} + \lambda_4 \mathcal{L}_{cont} + \lambda_5 \mathcal{L}_{text}, \tag{5}$$

Table 1: **Quantitative comparisons.** "†", "‡" and "⋆" denote reconstruction-only, understanding-only, and simultaneous scene understanding and 3D reconstruction methods, respectively. "-" indicates that the corresponding method do not support the corresponding task. "R→U" and "U→R" denote our Multi-View Mask Aggregation and Mask-Guided Geometry Refinement modules, respectively.

| | 3D Reconstruction | | | | | Scene Understanding | | | | | | | |
| | Depth Estimation | | Novel View Synthesis | | | Context Views (2D-only) | | | | Novel Views (3D-aware) | | | |
| | AbsRel↓ | RMSE↓ | PSNR↑ | SSIM↑ | LPIPS↓ | mIoU$_s$↑ | mAP↑ | PQ↑ | mIoU$_t$↑ | mIoU$_s$↑ | mAP↑ | PQ↑ | mIoU$_t$↑ |
|---|---|---|---|---|---|---|---|---|---|---|---|---|---|
| † pixelSplat[29] | 0.1812 | 0.4106 | 24.93 | 0.8065 | 0.2003 | - | - | - | - | - | - | | |
| † MVSplat[30] | 0.1697 | 0.3923 | 23.80 | 0.7871 | 0.2284 | - | - | - | - | - | - | | |
| † NoPoSplat[37] | 0.0944 | 0.2434 | 25.91 | 0.8147 | 0.1878 | - | - | - | - | - | - | | |
| ‡ Mask2former[39] | - | - | - | - | - | 0.5466 | 0.2486 | 0.6071 | - | - | - | - | - |
| ‡ LSeg[15] | - | - | - | - | - | 0.2601 | - | - | 0.2127 | - | - | - | - |
| ⋆ LSM[16] | 0.07468 | 0.2190 | 21.88 | 0.7336 | 0.3035 | 0.2745 | - | - | 0.1925 | 0.2707 | - | - | 0.1905 |
| ⋆ **Ours w/o R→U** | **0.07421** | **0.2081** | **25.96** | **0.8220** | **0.1841** | 0.5512 | 0.2529 | 0.6123 | 0.4572 | - | - | - | - |
| ⋆ **Ours w/o U→R** | 0.09619 | 0.2414 | 25.51 | 0.8168 | 0.1951 | 0.5893 | 0.2636 | 0.6569 | 0.5125 | 0.5875 | 0.2527 | 0.6456 | 0.5245 |
| ⋆ **Ours** | **0.07421** | **0.2081** | **25.96** | **0.8220** | **0.1841** | **0.5922** | **0.2817** | **0.6612** | **0.5273** | **0.5920** | **0.2714** | **0.6495** | **0.5270** |

where $I$ and $\hat{I}$ are rasterized and ground truth images, $\mathcal{L}_{mask}$ is derived from [39, 55], $\mathcal{L}_{mask} = \lambda_{ce}\mathcal{L}_{ce} + \lambda_{dice}\mathcal{L}_{dice} + \lambda_{cls}\mathcal{L}_{cls}$, where $\mathcal{L}_{ce}$ is binary cross-entropy loss, $\mathcal{L}_{dice}$ is dice loss and $\mathcal{L}_{cls}$ is classification loss. We follow [39, 55] set $\lambda_{ce}$, $\lambda_{dice}$ and $\lambda_{cls}$ to 5.0, 5.0, 2.0. In our training, we leverage both photometric loss and segmentation loss to simultaneously supervise 3D reconstruction and understanding. We set $\lambda_1, \lambda_2, \lambda_3, \lambda_4, \lambda_5$ to 1, 0.5, 0.05, 0.05, 1, respectively.

# 4 Experiments

## 4.1 Experimental Setup

**Implementation Details.** We utilize ScanNet[17] for training and validation, the largest public dataset that concurrently provides multi-view images with dense semantic/instance segmentation labels and text-referred segmentation labels[56]. We conduct training on 8 NVIDIA GeForce RTX 4090 GPUs, with our model trained for 100 epochs using a per-GPU batch size of 3 (total batch size of 24) for about 2 hours. AdamW optimizer[57] is employed with an initial learning rate of 1e-4 followed by cosine decay scheduling.

**Baselines.** Our experiments encompass not only isolated evaluation on separate tasks of 3D reconstruction and understanding, but also integrated evaluation on simultaneous understanding and 3D reconstruction. Therefore, we evaluate our method against three types of baseline methods, all of which are state-of-the-arts on their respective tasks: **1)** Sparse-view 3D reconstruction: pixelSplat[29], MVSplat[30], NoPoSplat[37]; **2)** Scene understanding: Mask2Former[39], LSeg[15]; **3)** Simultaneous scene understanding and 3D reconstruction: LSM[16]. All baseline methods are evaluated on ScanNet dataset under the same protocols as ours for fair comparison. To be more specific, for pixelSplat, MVSplat, NoPoSplat, Mask2Former, and LSM, we re-train their models following the training protocols of their official implementations using the processed ScanNet dataset same as ours. For reconstruction-only methods (i.e., pixelSplat, MVSplat, NoPoSplat), we only use their original rendering losses for supervision. For understanding-only method (i.e., Mask2Former), we only use its mask losses for supervision. For LSeg, since it already possesses general vision-language understanding capabilities, we directly adopt its pre-trained weights for evaluation.

**Metrics.** For 3D reconstruction, we evaluate the performance from two aspects: depth estimation and novel view synthesis, using depth accuracy metrics (i.e., AbsRel and RMSE) and image quality metrics (i.e., PSNR, SSIM and LPIPS), respectively. For scene understanding, we employ distinct evaluation protocols for 2D-based and 3D-based approaches. Specifically, without reconstructed 3D structures, 2D-based methods can only perform segmentation on the input context views. Therefore, we conduct "2D-only" evaluation on context view segmentation for 2D-based methods. In contrast, 3D-based approaches to simultaneous understanding and 3D reconstruction can perform 3D-level segmentation on the reconstructed 3D structures (e.g., 3D Gaussians in LSM and our method). Thus, for these methods, we leverage their characteristic of 3D-to-2D rasterization to project 3D-level segmentation results onto 2D masks, and conduct "3D-aware" evaluation on novel views. Note that our adoption of such evaluation on novel views is due to the lack of ground-truth segmentation labels for reconstructed 3D structures, which vary across different methods even for the same scene. As for different understanding tasks, we employ mIoU for semantic and text-referred segmentation, mAP for instance segmentation, and PQ for panoptic segmentation, where all metrics are computed with

global IDs as ground truths to penalize inconsistencies across views. To differentiate the metrics for semantic and text-referred tasks, we denote them as "mIoU$_s$" and "mIoU$_t$" in Table 1, respectively.

## 4.2 Main Results

**Quantitative Results.** As shown in Table 1, our approach outperforms all baselines across all tasks by a clear margin. For 3D reconstruction, unlike MVSplat and PixelSplat that require camera poses as input, or LSM that relies on ground-truth depth supervision, our framework eliminates dependency on pose and depth priors while achieving superior geometric accuracy and novel view synthesis quality. For scene understanding, existing methods like LSeg and Mask2Former are limited to 2D-only understanding of input context views and specific segmentation tasks. The only method

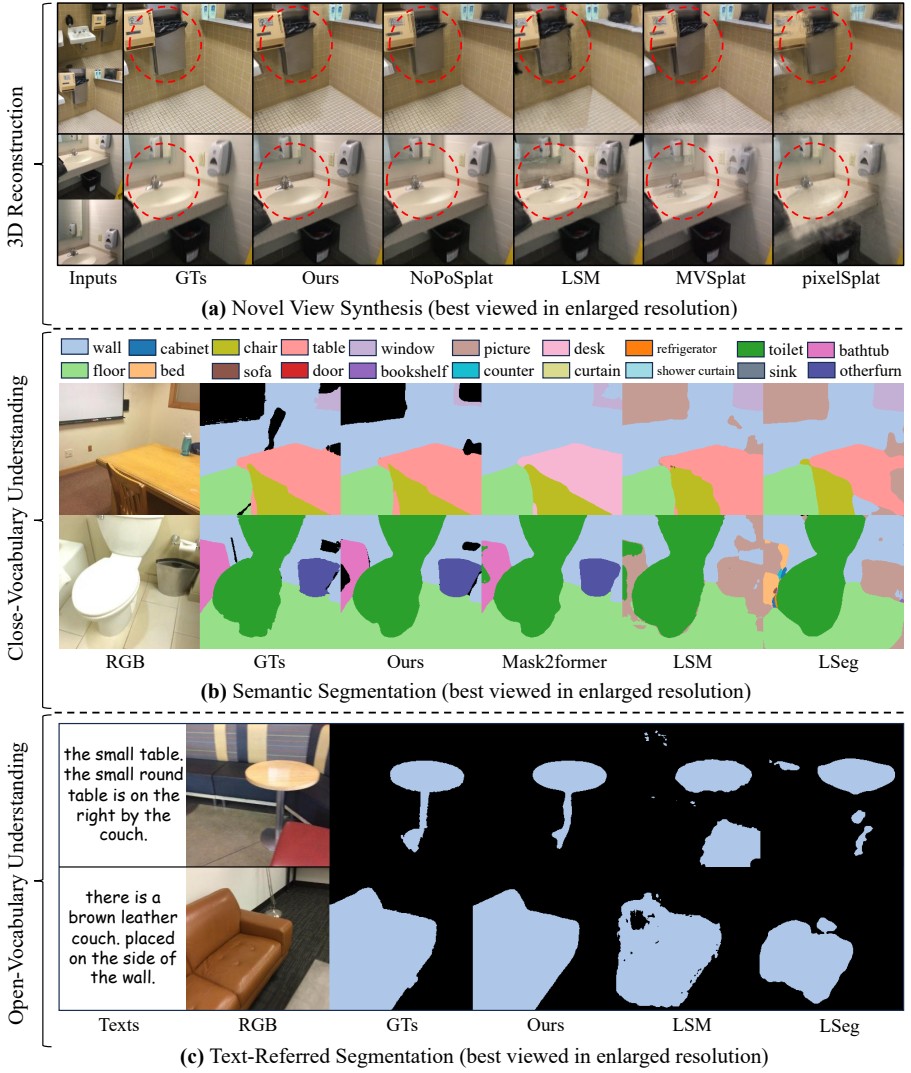

(a) Novel View Synthesis (best viewed in enlarged resolution)

(b) Semantic Segmentation (best viewed in enlarged resolution)

(c) Text-Referred Segmentation (best viewed in enlarged resolution)

Figure 5: **Qualitative Results.**

that can achieve 3D-aware understanding is LSM. However, its understanding capability is restricted by its source 2D model (LSeg) due to the nature of its feature alignment paradigm. Therefore, LSM can only support semantic and text-referred segmentation same as LSeg. Benefiting from our alignment-free paradigm and simultaneous task modeling, our generalist model supports both 2D and 3D understanding on comprehensive tasks including semantic, instance, panoptic, and text-referred segmentation within a unified framework, significantly exceeding other methods across all metrics. We also conduct experiments to validate the generalizability of our method to more input views, unseen data domains and real-world scenarios. ***Please refer to our appendices*** for more results.

Table 2: **Comparisons between design choices of Multi-View Aggregation.**

| | 3D Reconstruction | | | Context Views(2D-Only) | | | | Novel View (3D-aware) | | | | Memory |
|---|---|---|---|---|---|---|---|---|---|---|---|---|
| | PSNR↑ | SSIM↑ | LPIPS↓ | mIoU$_s$↑ | mAP↑ | PQ↑ | mIoU$_t$↑ | mIoU$_s$↑ | mAP↑ | PQ↑ | mIoU$_t$↑ | VRAM↓ |
| Early Aggregate w/o train. | **25.96** | **0.8220** | **0.1841** | 0.1126 | 0.051 | 0.2314 | 0.0012 | 0.1015 | 0.031 | 0.1765 | 0.0016 | ∼**23GB** |
| Early Aggregate w/ train. | 25.62 | 0.8153 | 0.1961 | 0.5442 | 0.2393 | 0.6011 | 0.4025 | 0.5434 | 0.2292 | 0.5849 | 0.3978 | ∼33GB |
| Ours | **25.96** | **0.8220** | **0.1841** | **0.5922** | **0.2817** | **0.6612** | **0.5273** | **0.5920** | **0.2714** | **0.6495** | **0.5270** | ∼**23GB** |

**Qualitative Results.** The qualitative results also demonstrate the superiority of our method. As illustrated in Fig.5 (a), thanks to our Mask-Guided Geometry Refinement module that improves reconstructed 3D geometries, our method exhibits better visual quality and fewer artifacts than others in novel view synthesis. As demonstrated in Fig.5 (b), thanks to our simultaneous task modeling and Multi-View Mask Aggregation mechanism, our method can effectively leverage geometric clues to improve understanding, and thus outperform 2D-only methods like Mask2Former with less erroneous inclusions in semantic masks. Besides, since our alignment-free framework decouples us from off-the-shelf 2D vision-language models, the quality of our segmentation masks is much better than methods like LSM and LSeg that rely on language-based querying for segmentation. Similar effects can also be observed in text-referred segmentation results (Fig.5 (c)), where our methods significantly surpasses LSM and LSeg with sharper boundaries and less fragments. ***Please refer to our appendices*** for more results (3D instance and panoptic segmentation, extension to versatile 3D editing, comparisons with more 3D-based baselines, real-world scenarios, etc).

## 4.3 Ablation Studies

**Reconstruction Helps Understanding (R→U).** We conduct ablations on our Multi-View Mask Aggregation module (denoted as "R→U" in Table 1). We can see that this module can significantly

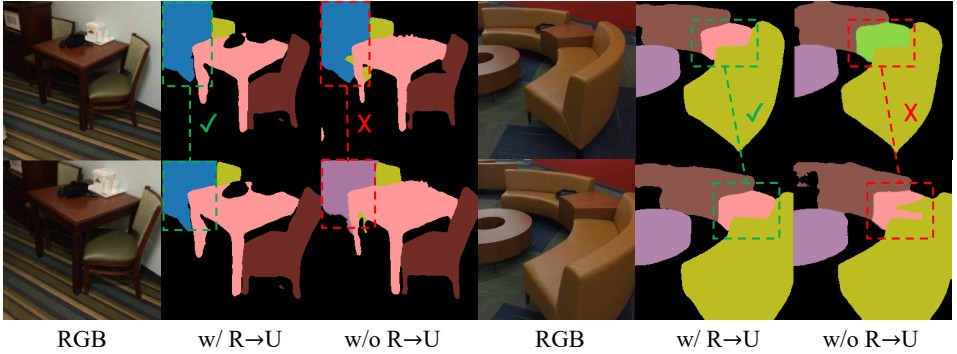

| RGB | w/ R→U | w/o R→U | RGB | w/ R→U | w/o R→U |

Figure 6: **Ablation on Multi-View Mask Aggregation (R→U).**

improve our performance in both 2D-only and 3D-aware scene understanding, without sacrificing 3D reconstruction accuracy due to its training-free nature. We attribute the improvement to our utilization of reconstructed 3D Gaussians as geometric clues, which can effectively enhance cross-view mask consistency as shown in Fig.6. We also note that, without this module to predict novel-view masks via 3D Gaussian rasterization, we can not enable 3D-aware tasks as denoted by "-" in Table 1.

**Understanding Helps Reconstruction (U→R).** We also conduct ablations on our Mask-Guided Geometry Refinement module (denoted as "U→R" in Table 1). With this module that employs mask guidance for geometry refinement, we can obtain much better 3D geometries ("depth estimation") and higher visual quality ("novel view synthesis") during reconstruction, which in turn enhances our performance in "Scene Understanding" thanks to our simultaneous modeling of the two tasks.

**Design Choices of Multi-View Aggregation.** As discussed in Sec.3.3, we can also implement multi-view aggregation via feature rasterization at the early feature encoding stage rather than mask decoding stage. However, as shown in Table 2, such early aggregation leads to poor performance without re-training our model. To incorporate it into training, we have to compress features to lower dimensions to avoid memory exhaustion. Nevertheless, the performance is still far inferior to our original design, while incurs significant training-time memory overhead due to the memory-intensive feature rasterization. We attribute this to the low compatibility between 3D Gaussians and compressed

2D features, which inevitably causes semantic information loss during feature rasterization, and in turn has a negative impact on the intricate forward process of our Unified Query Decoder.

## 5 Conclusion

We have introduced SIU3R, the first alignment-free framework for generalizable simultaneous understanding and 3D reconstruction. SIU3R unifies multiple understanding tasks into a set of unified learnable queries, which enables native 3D understanding without the need of alignment with 2D models. Two lightweight modules further bring significant mutual benefits between two tasks. Extensive experiments demonstrate the superiority of SIU3R to previous state-of-the-arts on both tasks, as well as its various applications including high fidelity 3D reconstruction, real-time native 3D understanding and versatile 3D editing.

**Limitations.** Currently, our SIU3R model is only trained on limited data compared to methods such as DUSt3R[34] and VGGT[58], which hinders its generalizability to broader visual domains. Additionally, since all the cross-view mask annotations in ScanNet dataset are obtained by projecting noisy 3D point clouds onto 2D images, the quality of our ground-truth labels is relatively poor, which may have a negative impact on our segmentation accuracy. We hope to address this in the future by introducing higher-quality labels or employing self-supervised methods less dependent on labeled data.

**Acknowledgements**. This work was supported in part by NSFC under Grant 62202389, in part by a grant from the Westlake University-Muyuan Joint Research Institute, in part by the Westlake Education Foundation, in part by the Headquarters Management Science and Technology Project of State Grid Corporation of China (No. 52090025001L-170-ZN), in part by the Tianjin Science and Technology Plan Project (No. 24YDPYSN00150), and in part by the Open Project Program of the State Key Laboratory of CAD&CG (Grant No. A2513), Zhejiang University.

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

## Appendix

In this appendix, we provide additional content to complement the main manuscript:

- Appendix A: Additional Implementation Details
- Appendix B: Comparisons with Per-Scene Optimization Methods
- Appendix C: Extend Our Model to Multi-View Inputs
- Appendix D: Comparison with Other Methods (e.g., DUSt3R, MASt3R, and VGGT)
- Appendix E: Additional Visualizations

## A    Additional Implementation Details

### A.1    Data Preprocessing

As described in Sec.4.1 of our main manuscript, we utilize ScanNet[17] for training and validation. We adopt the official training and validation dataset splitting of ScanNet, and then resize and crop original images to centered images at $256 \times 256$ resolution. The camera's intrinsic parameters have also been adjusted accordingly. We followed [37]'s camera conventions, where intrinsics are normalized and extrinsic parameters are OpenCV-style camera-to-world matrices.

Our data samples are obtained by randomly sampling context image pairs with certain overlaps. The overlap is determined by a pair-wise Intersection over Union (IoU) metric as shown in Fig.I. During training, we constrain the IoU to $[0.3, 0.8]$ to randomly select our training samples from scenes. Specifically, the IoU metric can be calculated as follows:

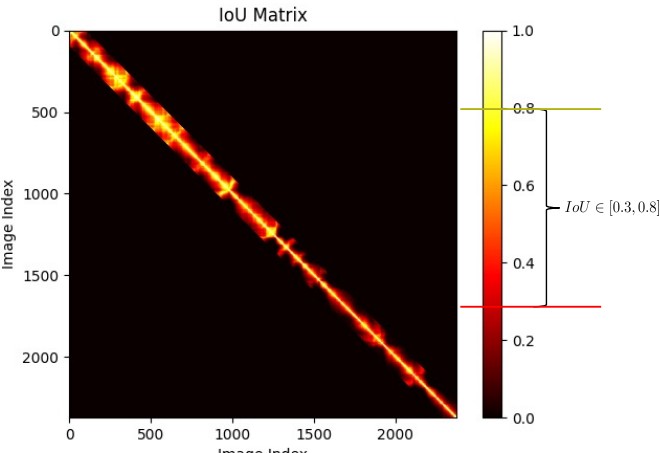

Figure I: **IoU matrix of ScanNet scene0011_00**

1. For a pair of images $I_1, I_2$, obtain their depths $D_1, D_2$, poses $P_1, P_2$ and intrinsic $K$.
2. Unproject $D_1$ into world coordinates and project them to $D_2$'s camera to obtain $D_1'$. Only depths satisfying $|D_1' - D_2| < 0.1$ are considered as valid.
3. Calculate the intersection over union ratio as:

$$IoU_{i \to j} = \frac{\#\text{valid projected depths}}{\#\text{total depths}}$$

4. Calculate $IoU_{1 \to 2}$ and $IoU_{2 \to 1}$.
5. Define the final IoU as:

$$IoU = \frac{IoU_{1 \to 2} + IoU_{2 \to 1}}{2}$$

The same IoU-based sampling strategy is also adopted in our evaluation, where we select 1,860 context image pairs to formulate the validation set. The curated evaluation benchmark and its processing scripts will be made publicly available for reproducing our results.

## A.2 Network Architecture and Hyperparameters

In Table I (a), the order from top to bottom are the network details of Image Encoder, Gaussian Decoder, Unified Query Decoder, respectively. In Table I (b), we specify loss weights for Eq.5 in our main manuscript, which is followed by parameters used in our training phase. To enable the Unified Query Decoder to leverage MASt3R features for scene understanding, we pre-trained the decoder on COCO dataset [59] while keeping the Image Encoder's weights frozen. The pre-trained weights will be publicly released to facilitate further research and development.

| (a) Network Architecture | | |
|---|---|---|
| Image Encoder | architecture | ViT encoder with Adapter[52] |
| | initialization | MASt3R[35] |
| | # depth of ViT encoder | 24 |
| | # embed dim of ViT encoder | 1024 |
| | # attn heads of ViT encoder | 16 |
| | positional embedding | RoPE |
| | # patchsize | 16 |
| | # interaction blocks of adapter | [5, 11, 17, 23] |
| | attention of adapter | MSDeformAttn |
| | # attention heads of adapter | 16 |
| | # inplanes of adapter spatial prior module | 64 |
| | # embed dim of adapter spatial prior module | 1024 |
| | # ref points | 4 |
| | # deform ratio | 0.5 |
| Gaussian Decoder | architecture | ViT decoder with DPT head[54] |
| | initialization | MASt3R[35] |
| | # depth of ViT decoder | 12 |
| | # embed dim of ViT decoder | 768 |
| | # attn heads of ViT decoder | 12 |
| | # channels of DPT head | 83 |
| | # sh degree | 4 |
| | # min gaussian scale | 0.5 |
| | # max gaussian scale | 15.0 |
| Unified Query Decoder | architecture | mask decoder[38, 39] |
| | # queries | 100 |
| | # probability score threshold of queries $\tau_c$ | 0.5 |
| | # probability score threshold of pixels $\tau$ | 0.3 |
| | # attn layers for text refer segmentation | 6 |
| (b) Hyperparameters | | |
| Loss Weights | # $\lambda_1, \lambda_2, \lambda_3, \lambda_4, \lambda_5$ | 1.0, 0.5, 0.05, 0.05, 1 |
| Training Details | learning rate scheduler | Cosine |
| | # epochs | 100 |
| | # learning rate | 1e-4 |
| | # batch size on each device | 3 |
| | training devices | 8 * RTX 4090 |
| | optimizer | AdamW[57] |
| | # beta1, beta2 | 0.9, 0.95 |
| | # weight decay | 0.05 |
| | # warm-up epochs | 3 |
| | # gradient clip | 1.0 |

Table I: **Details of network architecture and hyperparameters.** In the table, "#" denotes numerical parameters. We present parameters that specify our network architecture, and parameters used in our loss functions and training phase, in (a) and (b), respectively.

## A.3 Implementation Details about Versatile 3D Editing

As shown in Fig.1 of main manuscript, our simultaneous modeling of scene understanding and 3D reconstruction enables diverse 3D scene manipulations through unified pixel-aligned representations, including instance removal, replacement, relocation, and recoloring.

Here we take instance removal as an example to derive the implementation of such 3D editing:

1. Conduct inference to obtain SIU3R field with pixel-aligned 3D masks $M$ and Gaussians $\mathcal{G}$.

2. Remove Gaussians for a specified instance (ID $= ins\_id$):

$$\mathcal{G}' = \mathcal{G} \setminus \{\boldsymbol{g}_{\boldsymbol{v}}^{\boldsymbol{ij}} | \boldsymbol{M}_{ins}^{v,ij} = ins\_id\}$$

3. The modified Gaussians $\mathcal{G}'$ are rendered into original context views to obtain images $\mathcal{I}'$, with an off-the-shelf diffusion-based inpainting model [60] applied to fill the removed regions while ensuring visual coherence.

4. Conduct inference once again and rebuild SIU3R field from $\mathcal{I}'$.

For other 3D editing tasks (i.e. instance replacement, relocation and recoloring), we adopt a similar approach powered by different diffusion models [61–63].

## B    Comparisons with Per-Scene Optimization Methods

We also compare our approaches to methods (i.e., Feature-3DGS[11] and NeRF-DFF[9]) that require dense view capturing and per-scene optimization. Both of the two per-scene optimization methods follow a feature alignment paradigm similar to the feed-forward method LSM[16], where their 3D understanding capabilities are powered by off-the-shelf 2D vision language models that can only support language-guided segmentation. To enable the training of Feature-3DGS and NeRF-DFF, we uniformly select dense views (i.e., ∼100 images) as input and conduct per-scene optimization for each scene to align 3DGS or NeRF field with 2D features via rasterization. As shown in Table II, our method surpasses all of the feature alignment-based approaches by a large margin in the task of scene understanding, no matter they perform reconstruction in per-scene optimization (Feature-3DGS and NeRF-DFF) or feed-forward (LSM) manner. Besides, benefiting from our align-free framework, our method can further enable instance-level understanding tasks such as instance and panoptic segmentation. Furthermore, our method is the fastest in reconstruction speed, significantly surpassing Feature-3DGS and NeRF-DFF, and leading ahead of LSM. Considering that Feature-3DGS and NeRF-DFF use much more training images than our method, our performance in novel view synthesis is acceptable while achieving the best depth accuracy owing to our mask-guided geometry refinement module. As shown in Fig.III, we also make qualitative comparisons with these feature alignment-based methods, where our method achieves superior mask quality and semantic coherence.

Table II: **Quantitative comparisons with per-scene optimization methods.**

|  | Depth Estimation | | Novel View Synthesis | | | Scene Understanding | | | Efficiency |
|---|---|---|---|---|---|---|---|---|---|
|  | AbsRel↓ | RMSE↓ | PSNR↑ | SSIM↑ | LPIPS↓ | mIoU↑ | mAP↑ | PQ↑ | Reconstruction Time↓ |
| Feature-3DGS[11] | 0.1546 | 0.3585 | **28.69** | **0.8893** | 0.2171 | 0.3965 | - | - | 145.30min |
| NeRF-DFF[9] | 0.1846 | 0.4151 | 20.12 | 0.6252 | 0.5136 | 0.3410 | - | - | 2.71min |
| LSM[16] | 0.07468 | 0.2190 | 21.88 | 0.7336 | 0.3035 | 0.2745 | - | - | 0.24s |
| Ours | **0.07421** | **0.2081** | 25.96 | 0.8220 | **0.1841** | **0.5922** | **0.2817** | **0.6612** | **0.13s** |

## C    Extend to Multi-View Inputs

Building upon the insights of some works on feed-forward multi-view reconstruction (e.g., NoPoSplat[37], VGGT[58]), we make minor modifications to our model to support more input views for simultaneous understanding and 3D reconstruction. Specifically, compared to the model with two views, where cross-attention is only performed between the tokens of the two views, for multiple views, we perform cross-attention between the tokens of each view and the concatenated tokens of all other views. This is then followed by our Gaussian decoder and Unified Query Decoder, which predict multi-view 3D Gaussians and masks, respectively. We re-trained four variant models with different number of input views (i.e., 2, 4, 6, 8 views). The quantitative results, shown in the table III, demonstrate that as the number of views increases, our model exhibits improved performance in not only reconstruction but also 3D-aware segmentation on novel views. We think this improvement is primarily due to the availability of additional view information, which reduces geometry uncertainty, and enhances the quality of 3D reconstruction. This, in turn, boosts 3D scene understanding performance, thanks to our mutual-benefit mechanism (i.e., R → U).

Table III: **Quantitative results with different numbers of input views.**

| #Views | InferTime ↓ | InferVRAM ↓ | TrainVRAM ↓ | AbsRel ↓ | RMSE ↓ | PSNR ↑ | SSIM ↑ | LPIPS ↓ | mIoU ↑ | mAP ↑ | PQ ↑ |
|--------|-------------|-------------|-------------|----------|--------|--------|--------|---------|--------|-------|------|
| 2 | 0.067s | ∼ 4G | ∼ 24G | 0.074 | 0.208 | 25.96 | 0.822 | 0.184 | 0.592 | 0.271 | 0.650 |
| 4 | 0.097s | ∼ 5G | ∼ 38.5G | 0.072 | 0.205 | 26.28 | 0.822 | 0.179 | 0.611 | 0.283 | 0.687 |
| 6 | 0.140s | ∼ 6G | ∼ 56.5G | 0.071 | 0.203 | 26.45 | 0.830 | 0.175 | 0.617 | 0.287 | 0.693 |
| 8 | 0.185s | ∼ 7G | ∼ 77.5G | 0.071 | 0.201 | 26.65 | 0.838 | 0.170 | 0.620 | 0.289 | 0.697 |

# D  Comparison with Other Methods

Since DUSt3R, MAST3R, and VGGT are only designed to reconstruct scenes as 3D points rather than 3D Gaussians / masks, we note that they cannot be evaluated for the tasks of novel view synthesis and multi-task understanding. The comparison results are shown in the table below.

As for reconstruction performance, although DUSt3R, MAST3R and VGGT used much more training data (DUSt3R: 9 datasets, MASt3R: 14 datasets, VGGT: 17 datasets) and ground-truth depths for explicit geometry supervision, our method is only slightly inferior to them in terms of depth accuracy. We think such performance gap is reasonable because our method learns reconstruction solely through the supervision of novel view synthesis (NVS) task and was trained only on a single dataset. Besides, it is worth noting that our method exhibits the best depth accuracy among all the methods that support the novel view synthesis task (NVS), which focuses more on the rendering quality rather than merely on the geometry quality.

As for computational efficiency, our method exhibits faster inference speed compared to DUSt3R, MAST3R, and VGGT. We attribute this to the adoption of complex point post-processing for DUSt3R and MASt3R, and the significantly larger model size for VGGT (ours: 650M, VGGT: 1.1B).

Table IV: **Quantitative comparison on depth estimation task.**

| Method | Representation | Task | Runtime ↓ | AbsRel ↓ | RMSE ↓ |
|--------|----------------|------|-----------|----------|--------|
| DUSt3R[34] | Point | Depth | 0.14s | 0.058 | 0.178 |
| MASt3R[35] | Point | Depth | 1.09s | 0.057 | 0.175 |
| VGGT[58] | Point | Depth | 0.17s | 0.052 | 0.174 |
| MVSplat[30] | 3DGS | Depth & NVS | 0.06s | 0.170 | 0.392 |
| NoPoSplat[37] | 3DGS | Depth & NVS | 0.06s | 0.094 | 0.243 |
| LSM[16] | 3DGS | Depth & NVS & Segm. | 0.19s | 0.075 | 0.219 |
| Ours | 3DGS | Depth & NVS & Segm. | 0.07s | 0.074 | 0.208 |

# E  Additional Visualizations

## E.1  Real-World Captured Data

We collect some multi-view images in real-world scenarios using hand-held cellphones to further validate our generalization capability. As shown in Fig.II, we observe that our zero-shot performance in real-world scenes is surprisingly good in both reconstruction and segmentation.

## E.2  Instance, Panoptic and Text-Referred Segmentation

In our main manuscript, we have included qualitative comparisons with other methods and demonstrated our superiority in semantic segmentation. Here we provide additional qualitative results of instance and panoptic segmentation for further demonstration. As shown in Fig.IV, compared to 2D-based method Mask2Former[39] that leads to noisy mask boundaries, our method exhibits significantly higher mask quality. As shown in Fig.IV, when performing panoptic segmentation, our method exhibits excellent mask consistency across different views and significantly outperforms other methods in mask quality. Similar effects can also be observed in text-referred segmentation results as shown in Fig.VI. We attribute the superiority of our method to the simultaneous modeling of scene understanding and 3D reconstruction, which effectively leverages 3D geometric clues to aggregate semantic information from different views, and propagates them back to the original views to ensure cross-view consistency and improve segmentation accuracy.

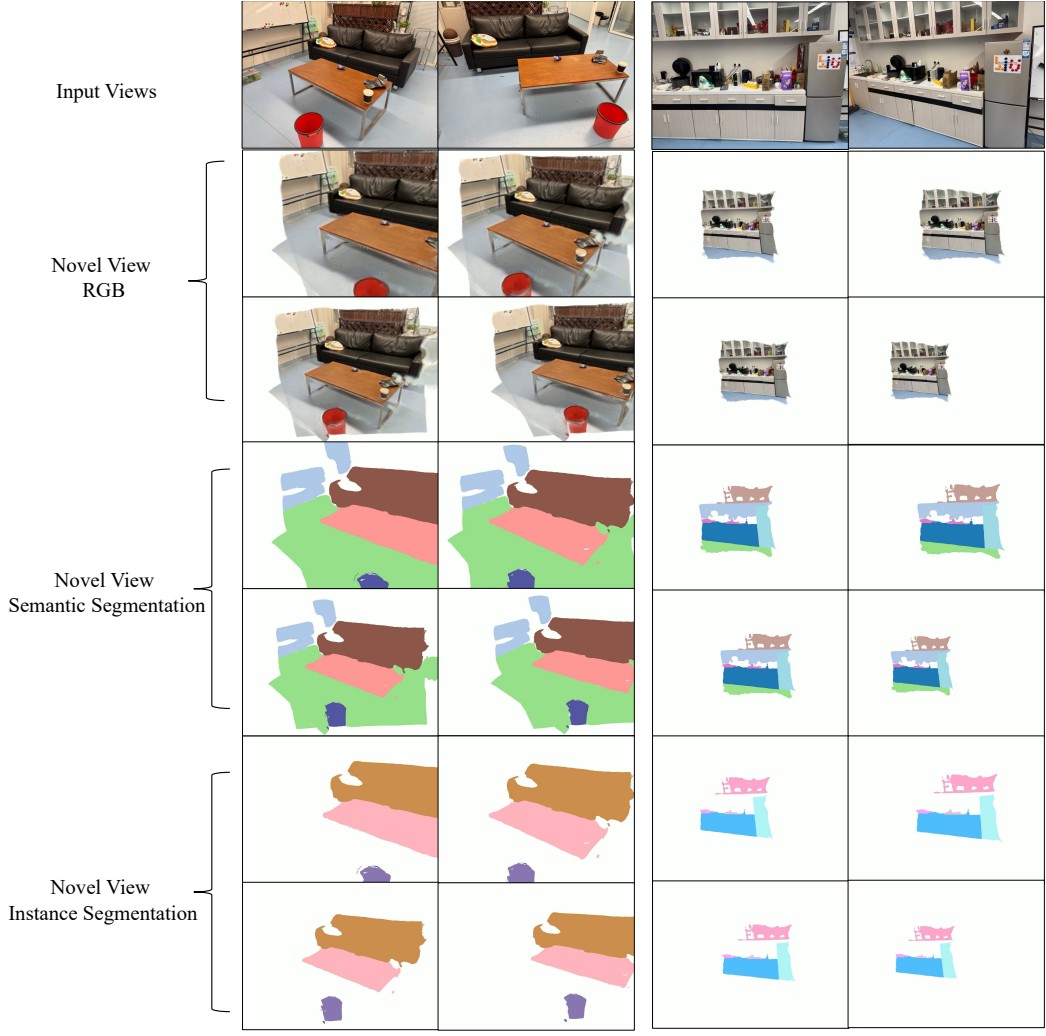

Figure II: **Qualitative result in real-world scenarios.**

## E.3 Depth Estimation

As illustrated in Fig.VII, compared to other feed-forward reconstruction methods, our approach achieves significantly superior depth quality with less artifacts and better coherence. We attribute this to our mask-guided geometry refinement module, which ensures geometry consistency within the same object instances under semantic guidance of 2D masks, and thus reduces erroneous depth variations that typically observed in other approaches.

## E.4 Versatile 3D Editing

As shown in Fig.VIII, we present a comprehensive set of versatile 3D editing results, demonstrating SIU3R's potential for diverse 3D manipulation applications. Furthermore, this capability establishes an effective baseline that bridges geometric reconstruction, scene understanding and manipulation in 3D environments.

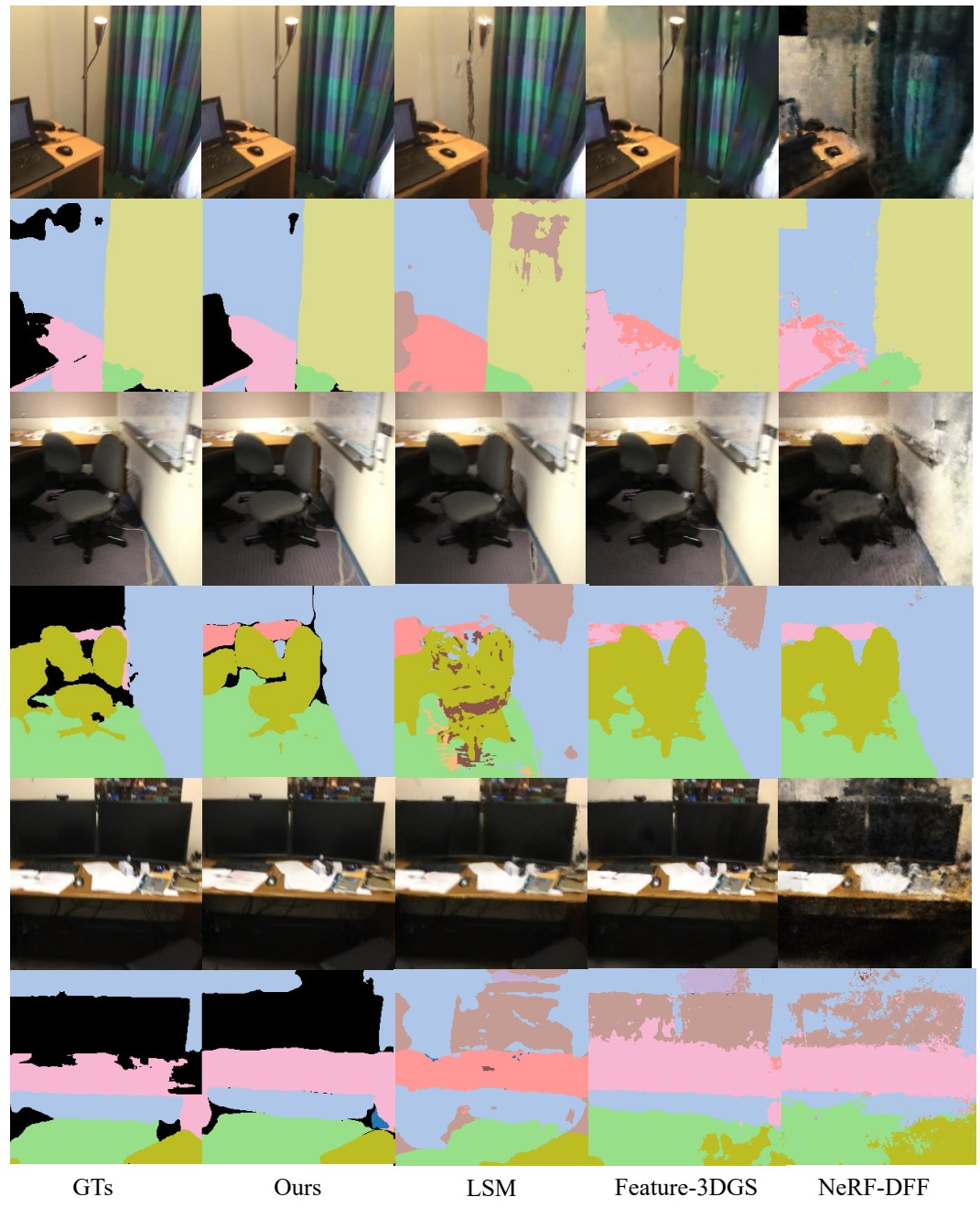

| GTs | Ours | LSM | Feature-3DGS | NeRF-DFF |

Figure III: **Qualitative comparisons with per-scene optimization methods.**

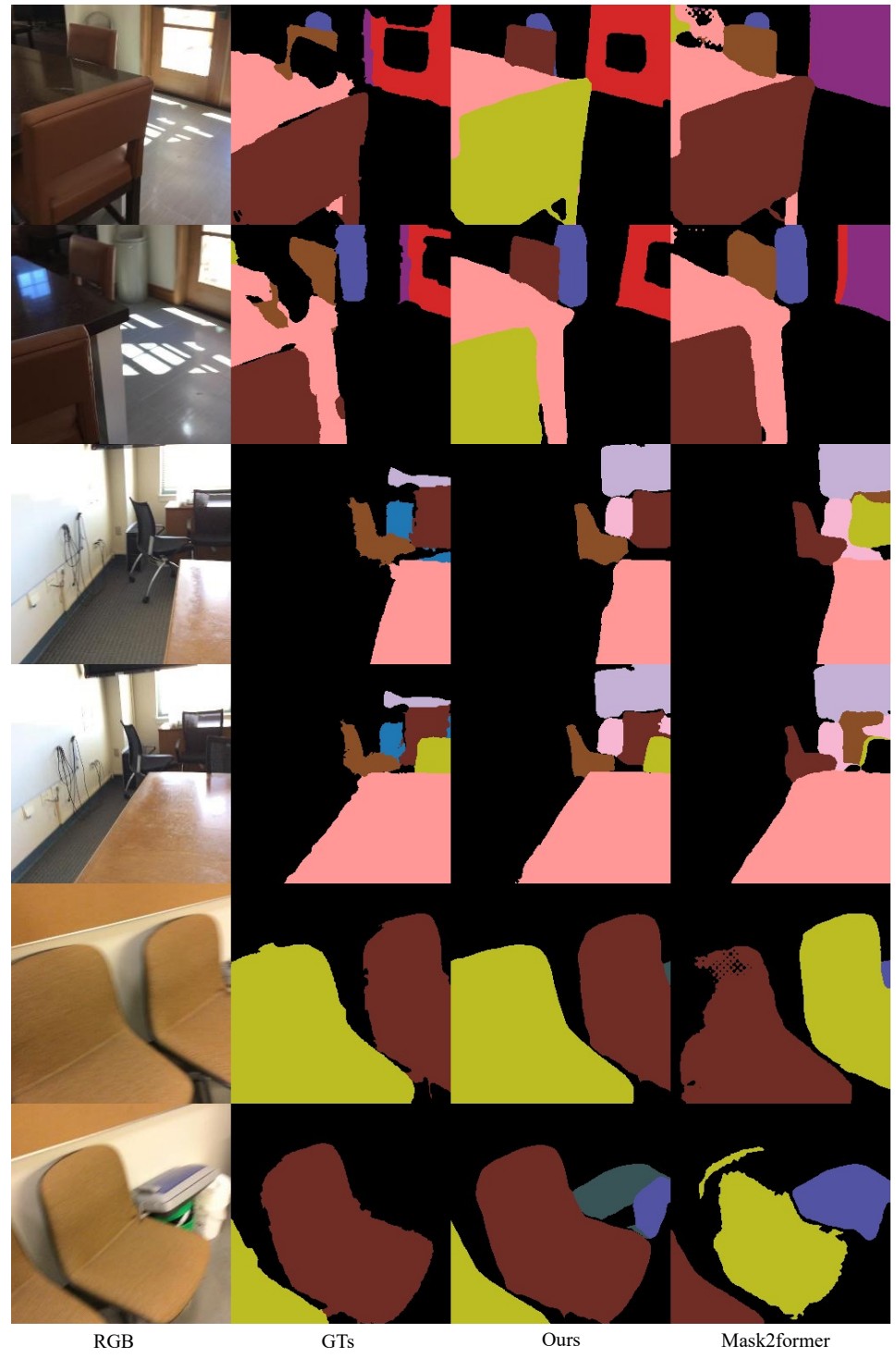

RGB          GTs          Ours          Mask2former

Figure IV: **Qualitative results of instance segmentation.**

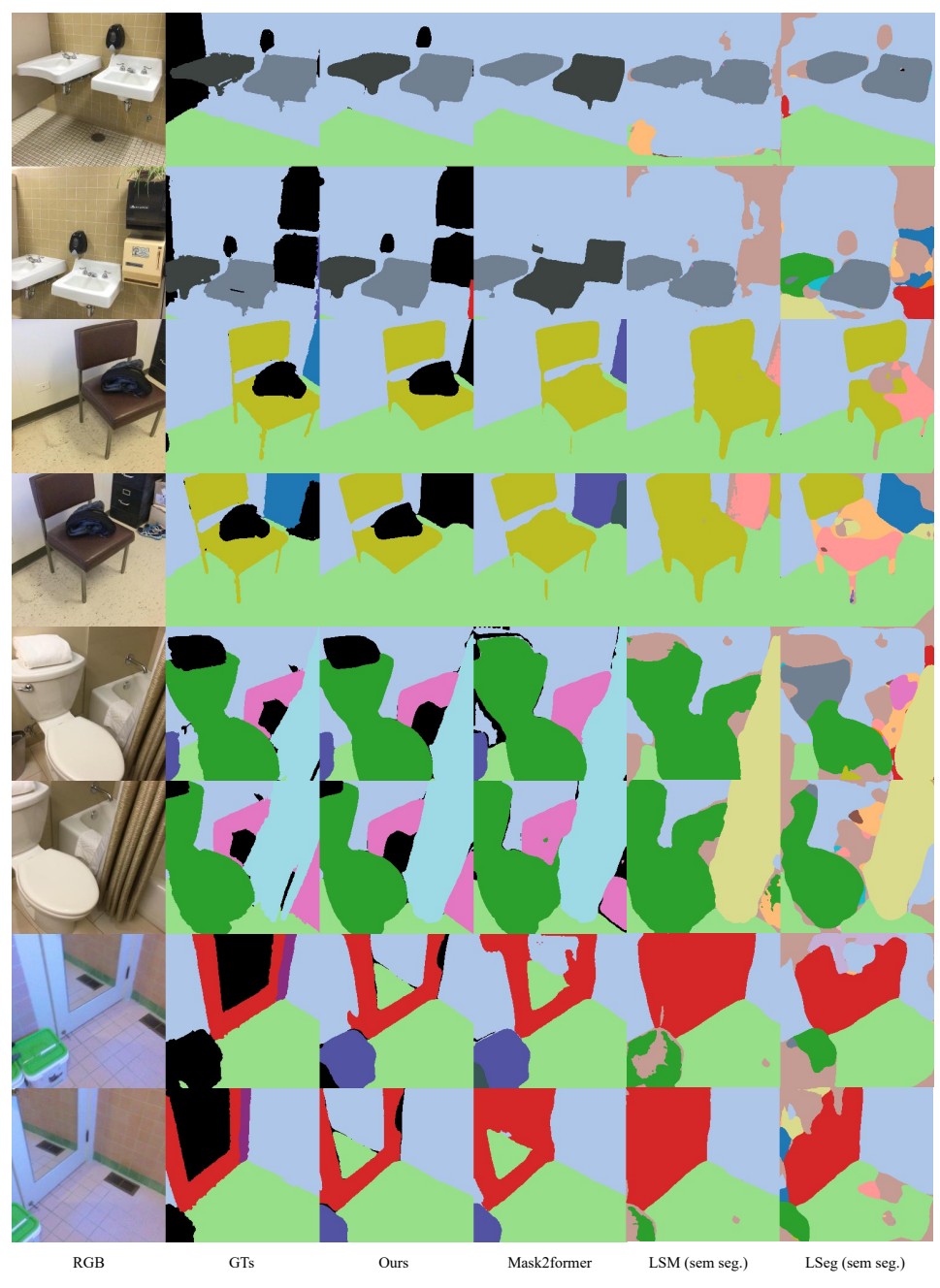

RGB       GTs       Ours       Mask2former       LSM (sem seg.)       LSeg (sem seg.)

Figure V: **Qualitative Results of Panoptic Segmentation.**

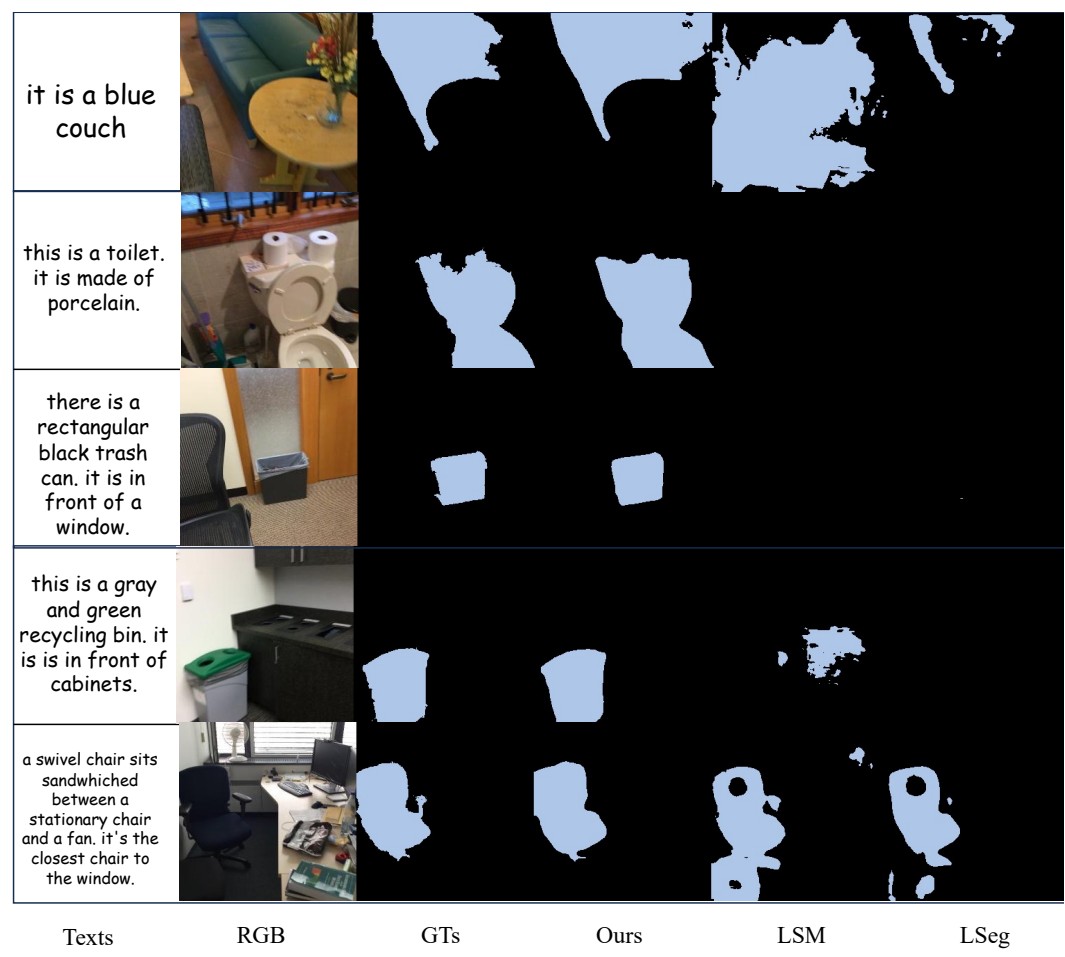

Figure VI: **Qualitative results of text-referred segmentation.**

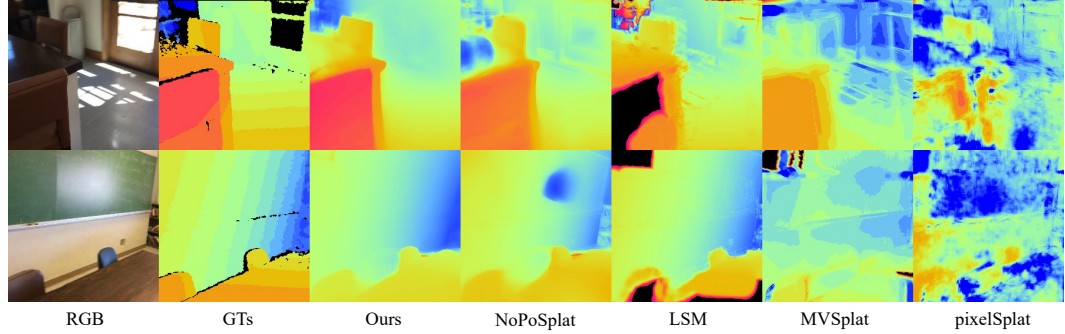

Figure VII: **Qualitative results of depth estimation.**

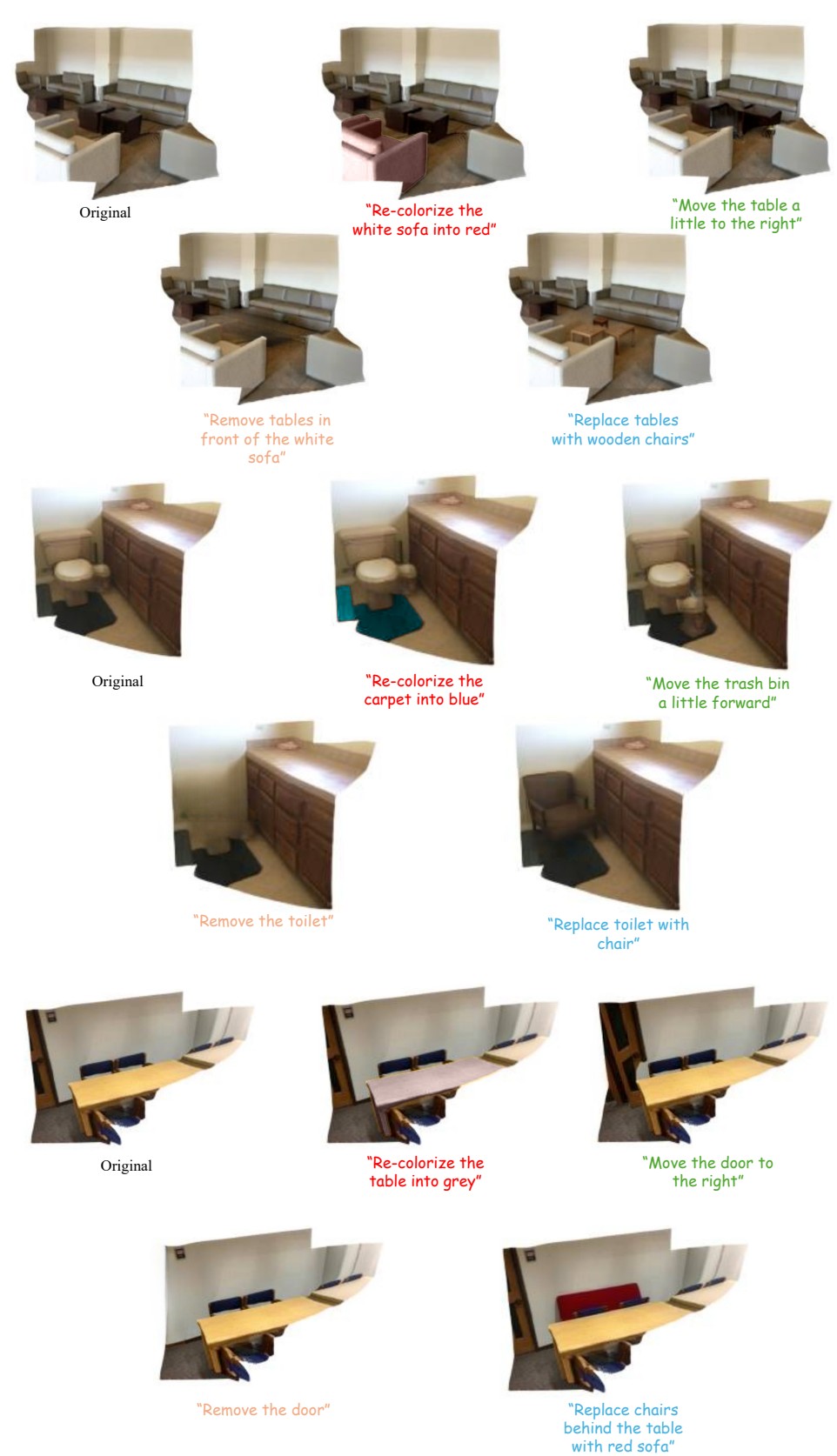

Original

"Re-colorize the
white sofa into red"

"Move the table a
little to the right"

"Remove tables in
front of the white
sofa"

"Replace tables
with wooden chairs"

Original

"Re-colorize the
carpet into blue"

"Move the trash bin
a little forward"

"Remove the toilet"

"Replace toilet with
chair"

Original

"Re-colorize the
table into grey"

"Move the door to
the right"

"Remove the door"

"Replace chairs
behind the table
with red sofa"

Figure VIII: **Qualitative results of versatile 3D editing.**

