# OpenReview forum: "SIU3R: Simultaneous Scene Understanding and 3D Reconstruction Beyond Feature Alignment"
_NeurIPS.cc/2025/Conference — NeurIPS 2025 spotlight_

### Official Review · Reviewer_Cqtb · 2025-06-08

**Clarity:** 3
**Significance:** 3
**Originality:** 3
**Rating:** 4
**Confidence:** 4

**Summary:**

This paper introduces SIU3R,  framework for simultaneous 3D reconstruction and scene understanding from a sparse pair of unposed images. The method proposes an "alignment-free" approach that bridges the two tasks using a shared, pixel-aligned 3D Gaussian representation. The Unified Query Decoder uses a single set of learnable queries to perform multiple understanding tasks (semantic and text-referred segmentation). The authors also introduce a "Mutual Benefit Mechanism," comprising two modules: one that uses 3D geometry to improve 2D mask consistency across views (Reconstruction -> Understanding) and another that uses 2D segmentation masks to refine the 3D geometry (Understanding -> Reconstruction). The authors demonstrate through experiments that their method achieves favorable results on both individual and combined tasks in the ScanNet dataset.

**Questions:**

- Could you provide a stronger motivation for the Unified Query Decoder? An ablation study comparing your approach to a simpler baseline—for instance, using an off-the-shelf 2D universal segmentation model (like Mask2Former) on each input view and then developing a method to lift these 2D masks to the 3D Gaussians—would be crucial to demonstrate the value of your proposed decoder.
- The L_cont loss (Eq. 4) seems to heavily rely on a local surface smoothness assumption. How does this loss function affect the reconstruction of objects with non-flat or intricate surfaces? To isolate the effectiveness of this heuristic, could you perform a controlled study where this loss is applied to a vanilla 3D Gaussian Splatting framework (i.e., not feed-forward) using ground-truth semantic masks? This would provide a clearer picture of its benefits and limitations.
- Could you please provide more detail on the implementation of this module? Specifically, Algorithm 1 suggests fusing multi-view semantics in 3D. How does the model aggregate class predictions and arrive at a single semantic label for each 3D Gaussian, particularly when faced with occlusions or conflicting predictions from different viewpoints?
- For the paper to be self-contained, could you please explicitly define the L_mask loss function in the main text or appendix, rather than only citing external work [38, 54]? A brief description of the objective would be sufficient.

**Ethical Concerns:**

["NO or VERY MINOR ethics concerns only"]

**Final Justification:**

The authors have adequately addressed my concerns. I have therefore updated my rating accordingly.

**Limitations:**

The authors have included a limitations section discussing the model's current constraint to two-view inputs and the impact of noisy ground-truth labels from the ScanNet dataset. This is a good start. However, the limitations I have raised in my review, such as the surface smoothness assumption in the L_cont loss and the lack of a clear mechanism for resolving multi-view conflicts in mask aggregation, are not addressed. Acknowledging these model-specific limitations would further improve the manuscript.

**Quality:**

3

**Strengths And Weaknesses:**

Strengths:

- The work addresses a challenging and highly relevant problem at the intersection of 3D vision and scene understanding. A generalizable, pose-free model that can produce both high-quality geometry and dense semantic labels has wide-ranging applications in robotics and augmented reality.
- The proposed "alignment-free" framework is  departure from recent methods that rely on distilling features from pre-trained 2D models. The idea of using unified queries to handle diverse 3D understanding tasks (semantic, instance, panoptic) within a single feed-forward pass is also an original and elegant contribution.

Weaknesses:

- The paper's core component for scene understanding, the Unified Query Decoder, is not sufficiently justified. It is a complex module, and it is not clear why it is preferable to a much simpler approach, such as running a state-of-the-art 2D segmentation model on each input view and lifting the results to 3D. The paper lacks a direct ablation study against such a baseline, making it difficult to assess the contribution of this specific design.
- The "Understanding Helps Reconstruction" (U->R) module relies on a continuity loss (L_cont) that encourages adjacent pixels within a predicted mask to have similar depths. This assumes that object surfaces are locally smooth or planar. This heuristic is likely to be ineffective or even detrimental for common objects with complex or curved geometries (e.g., chairs, bottles, cups), calling into question the general applicability of this component.
- Several key aspects of the method are not explained clearly. The process of Multi-View Mask Aggregation (R->U) is particularly opaque; Algorithm 1 is too high-level and does not explain how the model reaches a consensus on a semantic label for a 3D Gaussian when the input views provide conflicting information. Furthermore, the crucial segmentation loss L_mask is not defined in the paper, which simply refers to prior work, hindering the paper's self-containment and reproducibility.
- Insufficient Experimental Validation: The ablation studies are not convincing enough to support the claims about the "mutual benefit" modules. The benefits of U->R and R->U are not well-isolated from the rest of the proposed architecture. More controlled experiments, as suggested in the questions below, are needed to truly validate the effectiveness of these individual components.

---

> ### Author Rebuttal · Authors · 2025-07-27
>
> We thank reviewer Cqtb for recognizing our problem setting as well as the original and elegant contribution of our unified query design.
> Our detailed response to the feedback is as follows.
>
> **[W1,Q1] Stronger motivation for the Unified Query Decoder. Direct ablation study against a baseline that runs a state-of-the-art 2D segmentation model like Mask2Former and lifts the results to 3D.**
>
> **[Motivation]**
>
> Thanks for raising this concern.
> Our Unified Query decoder is built upon the contributions of 2D segmentation giants such as Mask2Former and SAM.
> However, different from previous 2D methods that can only handle single-image inputs or specific segmentation tasks, our Unified Query Decoder enables our model to accommodate multi-view inputs while ensuring multi-view mask consistency, as well as multi-task understanding including instance, semantic, panoptic, and text-referred segmentation.
>
> We note that these special designs **essentially serve the target of this work**: building a pixel-aligned representation-based simultaneous understanding and 3D reconstruction network, where the core is how to leverage pixel-aligned representation to combine multi-view masks and 3D Gaussians to achieve 3D-level understanding within a unified model.
>
> **Taking this into consideration, the motivation of Unified Query Decoder is twofold**: (1) Leverage unified queries shared across views to ensure multi-view mask consistency, which in turn ensures 3D-level segmentation accuracy. (2) Leverage unified queries shared across tasks to enable multi-task segmentation, which extends our approach to versatile 3D understanding applications.
> We promise we will include the above discussions in our revision.
>
> **[Ablation]**
>
> Thanks for the constructive suggestion.
> We implement two models: (1) the proposed method trained with only Gaussian Decoder (Reconstruction Only), (2) Mask2Former trained with multi-view masks (Segmentation Only). Both models are re-trained under the same settings as ours. Following the suggestion, we **directly lift 2D masks from Mask2Former to the 3D Gaussians**. We refer to this approach as "Mask2Former-GS".
> The comparison results are shown in the table below.
> We can see that **both efficiency and performance of Mask2Former-GS degrade compared to the original SIU3R model**.
>
> As for efficiency, the degradation of inference speed is caused by the extra computation from the additional image encoder of Mask2Former, highlighting the efficiency advantage of our unified model design.
>
> As for performance, we attribute the degradation of both reconstruction quality and segmentation accuracy to the lack of encoder feature sharing between Gaussian and mask decoders, indicating an interesting potential correlation between the reconstruction and segmentation tasks, which we leave to future work for further exploration.
>
> These observations further demonstrate our **"idea of using unified queries to handle diverse 3D understanding tasks within a single feed-forward pass is also an original and elegant contribution" as mentioned by the reviewer**.
> |Method|Efficiency|DepthEstimation|NovelViewSynthesis|Understanding(context-view)|Understanding(novel-view)|
> |:-|:-:|:-:|:-:|:-:|:-:|
> ||Runtime↓|AbsRel↓RMSE↓|PSNR↑LPIPS↓|mIoU↑mAP↑PQ↑|mIoU↑mAP↑PQ↑|
> |Mask2Former-GS| 0.15s |0.075\|0.209|25.74\|0.194|0.552\|0.250\|0.606|0.552\|0.249\|0.603|
> |Ours| **0.07s** |**0.074**\|**0.208**|**25.96**\|**0.184**|**0.592**\|**0.282**\|**0.661**|**0.592**\|**0.271**\|**0.650**|
>
>
> **[W2,Q2] The smooth heuristic of continuity loss is likely to be ineffective for objects with complex or curved geometries. Controlled study where the continuity loss is applied to a vanilla 3D Gaussian Splatting framework using ground-truth semantic masks?**
>
> **[Effectiveness]**
>
> Thanks for raising this concern. In our experiments, we only apply the mask-guided continuity loss to 3x3 neighbored pixels. We find **such small-range regularization rarely causes over-smoothness** in reconstructed 3D Gaussians, even for objects with complex or curved geometries.
> We have also prepared some qualitative results for demonstration, and found the visualized depths are decent for objects with complex or curved geometries. However, due to the updated rebuttal policy, we cannot provide the images here. **We promise we will include the results in our paper and code repository in revision.**
>
> **[Controlled Study]**
>
> Thanks for the constructive suggestion. We optimize vanilla 3D Gaussian splatting fields with and without mask-guided depth continuity loss for each scene in our validation set, and compare their depth and render quality in the table below.
> We can see that, with such intra-instance depth continuity constraint, **the vanilla 3DGS also exhibits better depth and render quality**, further demonstrate the effectiveness of our mask-guided depth continuity loss for per-scene optimization methods, validating the universality of our insight that "understanding helps reconstruction".
>
> |Method|DepthEstimation|NovelViewSynthesis|
> |:-|:-:|:-:|
> ||AbsRel↓RMSE↓|PSNR↑SSIM↑LPIPS↓|
> |3DGS w/o L_cont|0.155\|0.359|28.56\|0.872\|0.207|
> |3DGS w/ L_cont|**0.101**\|**0.261**|**28.81**\|**0.885**\|**0.191**|
>
>
> **[W3] Algorithm 1 does not explain how the model reaches a consensus on a semantic label for a 3D Gaussian when the input views provide conflicting information. Definition of the segmentation loss L_mask.**
>
> **[Algorithm 1]**
>
> Algorithm 1 primarily illustrates, at a pseudocode level, how to utilize multi-view 3D Gaussians to lift multi-view mask logits from 2D to 3D and achieve 3D-level multi-view aggregation based on rasterization.
> All operations correspond to PyTorch code and can be reproduced using the code in our supplementary materials.
>
> To explain how this process reaches a consensus on mask label for a 3D Gaussian when the input views provide conflicting information, **we will include the following description in our revision.**
>
> Specifically, before the "multi-view mask aggregation step" in Algorithm 1, different input views may exhibit conflicting information when decoding multi-view masks, which causes masks of the same object having different IDs across views. To address this, we reuse the predicted geometry-related 3D Gaussian parameters (i.e., mean, opacity, scale, rotation) to lift multi-view mask information (class-wise query probability maps) to the 3D space. **Through the alpha-blending mechanism of rasterization, pixels from different views but corresponding to the same object are explicitly forced to perform weighted averaging at the 3D level**, eliminating implicit 2D-level semantic inconsistencies before aggregation. Leveraging the aggregated mask information for mask decoding, we can further obtain multi-view consistent masks for 3D-level segmentation.
>
> **[L_mask]**
>
> The definition of the segmentation loss L_mask is as follows:
> $L_{mask}=\lambda_{bce} L_{bce} + \lambda_{dice} L_{dice} + \lambda_{ce} L_{ce} $. Specifically, $L_{bce}$ denotes binary cross-entropy loss that measures pixel-wise differences between the predicted foreground probability and ground truth. $L_{dice}$ denotes dice loss that measures the overlap ratio between the predicted foreground and ground truth. $L_{cls}$ denotes multi-class cross-entropy loss that measures differences between the predicted category probability and ground truth for each mask. In practice, we follow Mask2Former and set $\lambda_{bce},\lambda_{dice},\lambda_{cls}=5.0,5.0,2.0$. We will include this in our revision.
>
>
> **[W4] More controlled experiments on the R$\to$U and U$\to$R modules. They are not well-isolated from the rest of the proposed architecture.**
>
> Thanks for the constructive suggestion.
> We acknowledge that our R$\to$U and U$\to$R modules can indeed be used as plugins and integrated into other network architectures.
> Specifically, the R$\to$U module is a training-free module that can work as long as multi-view 3D Gaussians and mask logits are provided. The U$\to$R module is a model-free mask-guided loss function that can work as long as predicted / ground-truth multi-view masks are provided.
>
> To verify their effectivenesses under different network architecture, we implement a variant model without other components of our method (i.e., Unified Query Decoder), and directly combine this Gaussian-only network with Mask2Former ("Mask2Former-GS") as mentioned in [W1,Q1].
> We conduct ablation studies of R$\to$U and U$\to$R modules for this variant and show the results in the table below. We can see that **both of the two modules show similar improvements in reconstruction and understanding just as they do in the original architecture**.
> |Mask2Former-GS|DepthEstimation|NovelViewSynthesis|Understanding(context-view)|Understanding(novel-view)|
> |:-:|:-:|:-:|:-:|:-:|
> |R$\to$U\|U$\to$R|AbsRel↓RMSE↓|PSNR↑SSIM↑LPIPS↓|mIoU↑mAP↑PQ↑|mIoU↑mAP↑PQ↑|
> |✘\|✘|0.090\|0.233|25.02\|0.812\|0.202|0.543\|0.236\|0.587|N/A\|N/A\|N/A|
> |✓\|✘|0.090\|0.233|25.02\|0.812\|0.202|0.551\|0.247\|0.595|0.550\|0.243\|0.592|
> |✘\|✓|0.075\|0.209|25.74\|0.817\|0.194|0.545\|0.241\|0.590|N/A\|N/A\|N/A|
> |✓\|✓|**0.075**\|**0.209**|**25.74**\|**0.817**\|**0.194**|**0.552**\|**0.250**\|**0.606**|**0.552**\|**0.249**\|**0.603**|
>
>
> **[Q3] Could you please provide more details on the implementation of Algorithm 1?**
> Please refer to [W3] for details.
>
>
> **[Q4] Could you please explicitly define the L_mask loss function?**
> Please refer to [W3] for details.

---

> > ### Comment · Reviewer_Cqtb · 2025-08-05
> >
> > Thanks for the detailed response and additional results. The authors have adequately addressed my concerns. I have therefore raised my score.

---

> > > ### Author Response · Authors · 2025-08-05
> > >
> > > Thank you for your positive recommendation and helpful feedback. In the revised paper we will add the new results and the necessary clarifications.

---

### Official Review · Reviewer_qLPk · 2025-06-17

**Clarity:** 3
**Significance:** 3
**Originality:** 3
**Rating:** 4
**Confidence:** 4

**Summary:**

This paper introduces SIU3R, a novel, alignment-free framework for generalizable simultaneous 3D reconstruction and multi-task understanding. SIU3R bridges reconstruction and understanding tasks via pixel-aligned 3D representation, and unifies multiple understanding tasks into a set of unified learnable queries. Moreover, this paper further conducts in-depth analyses of the mutual benefits of reconstruction and understanding tasks and proposes two lightweight modules to facilitate their interaction.

**Questions:**

Based on the analysis above, there are two questions: (1) Given the limited evaluation on ScanNet, how does SIU3R perform on other diverse datasets and real-world scenarios to demonstrate its true generalization capabilities?   (2) How does SIU3R compare with state-of-the-art feed-forward methods like Dust3R, MAST3R, and VGGT in terms of reconstruction quality and computational efficiency?

**Ethical Concerns:**

["NO or VERY MINOR ethics concerns only"]

**Final Justification:**

The responses and additional experiments have addressed most of my concerns. While the generalization capability of the method in outdoor scenarios remains to be thoroughly validated, I understand the time constraints during the rebuttal period make it challenging to conduct more experiments. Therefore, I keep my initial score of borderline accept.

**Limitations:**

yes

**Paper Formatting Concerns:**

no formatting concerns

**Quality:**

3

**Strengths And Weaknesses:**

Strengths:
This paper introduces a novel alignment-free framework for generalizable simultaneous understanding and 3D reconstruction, which bridges reconstruction and understanding via pixel-aligned 2D-to-3D lifting, and unifies multiple 3D understanding tasks (i.e., semantic, instance, panoptic and text-referred segmentation) into a set of unified learnable queries. Moreover, this work conducts in-depth exploration of inter-task mutual benefits in the realm of simultaneous understanding and 3D reconstruction. The framework is capable of handling multiple tasks, and its methodological design demonstrates innovation. The experimental results are good.

Weaknesses:
1.  Training and testing solely on a single dataset (ScanNet) is insufficient to demonstrate good generalization capability and algorithm effectiveness. Would it be possible to train a generalized network that, similar to Dust3R, possesses strong generalization abilities across different scenarios?
2.  The paper compares performance on sparse-view 3D reconstruction tasks, but the baselines chosen are primarily methods that don't require extensive training periods, which may not constitute a fair comparison. Would it be more appropriate to compare with other feed-forward methods such as Dust3R, MAST3R, and VGGT?

---

> ### Author Rebuttal · Authors · 2025-07-27
>
> We thank reviewer qLPk for recognizing our novel alignment-free framework, in-depth exploration of mutual benefits, and good experimental results.
> Our detailed response to the feedback is as follows.
>
> **For your information**, in addition to addressing the concerns mentioned in this review, we have provided more experimental results (e.g., extend SIU3R to support more input views as suggested by reviewer oN8Y, oS3w) that may be of your interest.
>
> **[W1] Would it be possible to train a generalized network that, similar to DUSt3R, possesses strong generalization abilities across different scenarios?**
> Due to the limited time of rebuttal phase, we are unable to collect and process a large amount of datasets comparable in scale to DUSt3R (9 datasets) with both multi-view images and mask annotations.
> Therefore, we process two representative indoor multi-view datasets with different sensor characteristics (i.e., ScanNet++, Replica) as additional training / evaluation data.
> Specifically, we conducted two experiments, i.e. zero-shot generalization on ScanNet++ & Replica, and train a general model by treating ScanNet++ & Replica as additional training data. The detailed results are shown in table below.
>
> It can be seen that SIU3R has certain zero-shot generalization capability even it is trained solely with ScanNet. However, due to the size of training data (i.e. ScanNet), its generalization performance still has rooms to improve. Despite this, our method still exhibits better generalization performance compared to other baselines under the same training settings.
>
> To further verify if SIU3R has performance gains with more training data, we train it with more datasets to get a general model. The results demonstrate that SIU3R achieved improved performance even the training data exhibits different characteristics. It demonstrates that SIU3R should achieve good performance as the dataset size further increases, as what VGGT (a concurrent work as ours, from CVPR'25) demonstrates.
>
> ||$\qquad$ScanNet|$\qquad$ScanNet++|$\qquad$Replica|
> |:-|:-:|:-:|:-:|
> |**Method**| PNSR↑AbsRel↓mIoU↑| PNSR↑AbsRel↓mIoU↑| PNSR↑AbsRel↓mIoU↑|
> |pixelSplat (train on ScanNet only)|24.93\|0.181\|N/A|17.91\|0.274\|N/A|18.12\|0.286\|N/A|
> |MVSplat (train on ScanNet only)|23.80\|0.170\|N/A|18.27\|0.261\|N/A|18.43\|0.277\|N/A|
> |NoPoSplat (train on ScanNet only)|25.91\|0.094\|N/A|19.76\|0.107\|N/A|19.88\|0.110\|N/A|
> |Ours (train on ScanNet only)|25.96\|0.074\|0.592|20.05\|0.084\|0.482|20.46\|0.087\|0.376|
> |Ours (train on ScanNet, ScanNet++, Replica)|**26.06**\|**0.073**\|**0.607**|**23.39**\|**0.074**\|**0.498**|**23.16**\|**0.079**\|**0.492**|
>
> Besides, we collect some multi-view images in real-world scenarios using hand-held cellphones to further validate our generalization capability. We observe that our zero-shot performance in real-world scenes is surprisingly good in both reconstruction and segmentation, which is qualitatively on par with that on ScanNet dataset. Due to the updated rebuttal policy this year, we cannot provide the qualitative results here. **We promise we will include the results in our paper and code repository in revision.**
>
> **[W2] Compare with other feed-forward methods like DUSt3R, MAST3R, and VGGT.**
> Thanks for the constructive suggestion.
>
> Since DUSt3R, MAST3R, and VGGT are only designed to reconstruct scenes as 3D points rather than 3D Gaussians / masks, we note that they cannot be evaluated for the tasks of novel view synthesis and multi-task understanding.
> The comparison results are shown in the table below.
>
> **As for reconstruction performance**, although DUSt3R, MAST3R and VGGT used much more training data (DUSt3R: 9 datasets, MASt3R: 14 datasets, VGGT: 17 datasets) and ground-truth depths for explicit geometry supervision, our method is **only slightly inferior to them in terms of depth accuracy**.
> We think such performance gap is reasonable because our method learns reconstruction solely through the supervision of novel view synthesis (NVS) task and was trained only on a single dataset.
> Besides, it is worth noting that our method **exhibits the best depth accuracy among all the methods that support the novel view synthesis task (NVS)**, which focuses more on the rendering quality rather than merely on the geometry quality.
>
> **As for computational efficiency**, our method outperforms DUSt3R, MAST3R and VGGT by a clear margin.
> We attribute this to the adoption of complex point post-processing for DUSt3R and MASt3R, and the significantly larger model size for VGGT (ours: 650M, VGGT: 1.1B).
>
> We will include the results in our revision.
>
> |Method|Representation|$\qquad$$\quad$$\text{ }$$\text{ }$Task|Runtime $\downarrow$|AbsRel $\downarrow$|RMSE $\downarrow$|
> | :---: | :---: |:---:| :---: | :---: | :---: |
> |DUSt3R|Point|Depth|0.14s|0.058|0.178
> |MASt3R|Point|Depth|1.09s|0.057|0.175
> |VGGT|Point|Depth|0.17s|0.052|0.174
> |MVSplat|3DGS|Depth & NVS|0.06s|0.170|0.392
> |NoPoSplat|3DGS|Depth & NVS|0.06s|0.094|0.243
> |LSM|3DGS|Depth & NVS & Segm.|0.19s|0.075|0.219
> |Ours|3DGS|Depth & NVS & Segm.|0.07s|0.074|0.208
>
> **[Q1] How does SIU3R perform on other diverse datasets and real-world scenarios to demonstrate its true generalization capabilities?**
> Please refer to [W1] for details.
>
> **[Q2] How does SIU3R compare with DUSt3R, MAST3R, and VGGT in terms of reconstruction quality and computational efficiency?**
> Please refer to [W2] for details.

---

> > ### Comment · Reviewer_qLPk · 2025-08-04
> > **Official Comment by Reviewer qLPk**
> >
> > Thank you for your effort in rebuttal. The responses and additional experiments have addressed most of my concerns. While the generalization capability of the method in outdoor scenarios remains to be thoroughly validated, I understand the time constraints during the rebuttal period make it challenging to conduct more experiments. Therefore, I keep my initial score of borderline accept.

---

> > > ### Author Response · Authors · 2025-08-04
> > > **Response to Comment from Reviewer#qLPk**
> > >
> > > Thanks for your positive recommendation and the supportive feedback on our paper. We are glad that our responses have addressed most of your concerns.

---

### Official Review · Reviewer_oS3w · 2025-06-23

**Clarity:** 2
**Significance:** 2
**Originality:** 2
**Rating:** 4
**Confidence:** 2

**Summary:**

This paper introduces SIU3R, a framework for ​​simultaneous segmentation and 3D reconstruction​​ from two unposed images. SIU3R eliminates the need for 2D-to-3D feature alignment through an ​​alignment-free​​ approach leveraging ​​pixel-aligned 3D Gaussians. Specifically, there are two key components, including 1) a unified query decoder​​ that decodes multi-task (semantic, instance, panoptic, text-referred) masks from a single set of learnable queries shared across views; 2) a ​mutual benefit mechanism​​ comprising a) multi-view mask aggregation ​​ that uses reconstructed Gaussians to improve mask consistency across views without training, and b) ​​mask-guided geometry refinement that enforces intra-instance depth continuity to refine the geometry using predicted masks. Extensive experiments on ScanNet demonstrate advanced performance on individual tasks (3D reconstruction and segmentation) and validate the effectiveness of each component.

**Questions:**

1. In Equation 2, you use the queried text to search for the matching instance. If there are more than two instances of the same category appearing in the scene, will this solution also work?
2. The comparison with LSM on training efficiency is notable. What are the primary factors contributing to this efficiency?
3. What are the computational/memory bottlenecks when scaling beyond two views? Are there architectural modifications considered for efficient multi-view scaling?

**Ethical Concerns:**

["NO or VERY MINOR ethics concerns only"]

**Final Justification:**

This paper introduces SIU3R, a framework for simultaneous segmentation and 3D reconstruction using two unposed images. My initial concerns were primarily related to the limitations of the two-view setup, the evaluation on a single dataset, and the lack of an ablation study. The authors have effectively addressed most of these points in their rebuttal by providing additional experiments. I therefore recommend acceptance (borderline accept).

**Limitations:**

yes

**Paper Formatting Concerns:**

I have found no formatting issues in this paper.

**Quality:**

2

**Strengths And Weaknesses:**

**Strengths:**
1.  The proposed R→U (Multi-View Mask Aggregation) and U→R (Mask-Guided Geometry Refinement) modules are lightweight and intuitive and demonstrably improve both tasks bidirectionally. Table 1 clearly shows their positive impact.
2. Training time (2h on 8x4090) is significantly lower than competitors (e.g., LSM's 3 days on 8xA100). The pose-free paradigm enhances practicality.

**Weaknesses:**
1. The experimental validation is ​​exclusively conducted on ScanNet. While ScanNet is a standard benchmark for indoor 3D scene understanding and reconstruction, this single-dataset focus raises questions about the ​​generalizability​​ of SIU3R's results. Performance on ​​outdoor scenes​​ (e.g., KIT) or datasets with ​​different sensor characteristics​​ (e.g., Matterport3D) remains unverified.
2. The current framework is designed and evaluated exclusively on two-view inputs. The authors acknowledge this as a limitation for large-scale scene understanding. It is unclear how the architecture, particularly the unified query decoder, would scale computationally and effectively with a larger number of input views or video sequences.
3. The ablation studies (Sec. 4.3) effectively demonstrate the individual contribution of the R→U and U→R modules. ​​However, some important experiments are missing from this paper, 1) multitask training without both R -> U  and U -> R, and 2) independent training for each task. These experiments will help you better demonstrate the effectiveness of your method.
4. The paper ​​equates "scene understanding" solely with segmentation tasks​​ (semantic, instance, panoptic, text-referred segmentation). While segmentation is fundamental, ​​3D scene understanding​​ encompasses a broader spectrum, including ​​3D dense captioning​​, ​​3D visual grounding​​, and ​​3D question answering​​. SIU3R's capabilities and representations are not evaluated or discussed for these higher-level semantic tasks. This limits the claimed, "versatile 3D reconstruction, understanding, and editing applications" (Fig 1c, Abstract). Otherwise, I suggest the authors narrow down the claim and research scope made in this paper.

---

> ### Author Rebuttal · Authors · 2025-07-27
>
> We thank reviewer oS3w for recognizing our lightweight and intuitive designs, as well as our significance in training efficiency compared to LSM.
> Our detailed response to the feedback is as follows.
>
> **[W1] Performance on datasets with ​​different sensor characteristics remains unverified**
> To fully verify our performance on other datasets, we conducted two experiments, i.e. zero-shot generalization on datasets with different sensor characteristics (ScanNet++ & Replica), and train a general model with more datasets. The detailed results are shown in table below.
>
> It can be seen that SIU3R has certain zero-shot generalization capability even it is trained solely with ScanNet. However, due to the size of training data (i.e. ScanNet), its generalization performance still has rooms to improve. Despite this, our method still exhibits better generalization performance compared to other baselines under the same training settings.
>
> To further verify if SIU3R has performance gains with more training data, we train it with more datasets to get a general model. The results demonstrate that SIU3R achieved improved performance even the training data exhibits different characteristics. It demonstrates that SIU3R should achieve good performance as the dataset size further increases, as what VGGT (a concurrent work as ours, from CVPR'25) demonstrates.
>
> ||$\qquad$ScanNet|$\qquad$ScanNet++|$\qquad$Replica|
> |:-|:-:|:-:|:-:|
> |**Method**| PNSR↑AbsRel↓mIoU↑| PNSR↑AbsRel↓mIoU↑| PNSR↑AbsRel↓mIoU↑|
> |pixelSplat (train on ScanNet only)|24.93\|0.181\|N/A|17.91\|0.274\|N/A|18.12\|0.286\|N/A|
> |MVSplat (train on ScanNet only)|23.80\|0.170\|N/A|18.27\|0.261\|N/A|18.43\|0.277\|N/A|
> |NoPoSplat (train on ScanNet only)|25.91\|0.094\|N/A|19.76\|0.107\|N/A|19.88\|0.110\|N/A|
> |Ours (train on ScanNet only)|25.96\|0.074\|0.592|20.05\|0.084\|0.482|20.46\|0.087\|0.376|
> |Ours (train on ScanNet, ScanNet++, Replica)|**26.06**\|**0.073**\|**0.607**|**23.39**\|**0.074**\|**0.498**|**23.16**\|**0.079**\|**0.492**|
>
> **[W2, Q3] How the architecture would scale computationally and effectively with more views. Modifications for efficient multi-view scaling?**
> Thanks for raising this concern.
>
> Inspired by concurrent works on feed-forward multi-view reconstruction (e.g., VGGT), we **make minor modifications to our model and make it support more input views**.
> Specifically, compared to the model with two views, where cross-attention is only performed between the tokens of the two views, for multiple views, we perform cross-attention between the tokens of each view and the concatenated tokens of all other views. This is followed by our Gaussian decoder and Unified Query Decoder to predict multi-view 3D Gaussians and masks, respectively.
> We re-trained 4 variant models with different number of input views (i.e., 2, 4, 6, 8 views). The quantitative results are shown in the table below.
> We can see that, **as No. Views increases, our model achieves better performance in not only reconstruction but also 3D-aware segmentation on novel views**. The main reason is that, with more view information available to reduce geometry uncertainty, the quality of 3D reconstruction improves, which, thanks to the design of our R$\to$U module, in turn enhances the segmentation performance.
>
> We have also prepared some qualitative results of handling 8 views and larger scene scales. However, due to the updated rebuttal policy, we cannot provide here. **We promise we will include the results in our paper and code repository in revision.**
> |No.Views|InferTime|InferVRAM|TrainVRAM|AbsRel↓|RMSE↓|PSNR↑|SSIM↑|LPIPS↓|mIoU↑|mAP↑|PQ↑|
> |:-:|:-:|:-:|:-:|:-:|:-:|:-:|:-:|:-:|:-:|:-:|:-:|
> |2|0.067s|~4G|~24G|0.074|0.208|25.96|0.822|0.184|0.592|0.271|0.650|
> |4|0.097s|~5G|~38.5G|0.072|0.205|26.28|0.822|0.179|0.611|0.283|0.687|
> |6|0.140s|~6G|~56.5G|0.071|0.203|26.45|0.830|0.175|0.617|0.287|0.693|
> |8|0.185s|~7G|~77.5G|0.071|0.201|26.65|0.838|0.170|0.620|0.289|0.697|
>
> **As for efficient multi-view scaling**, we believe this remains an open question within the field. However, from an engineering perspective, we can utilize FlashAttention and efficient optimizer tools (e.g., DeepSpeed ZeRO-3) to enable efficient training.
>
> We also note that, at the time of our initial submission, most prior feed-forward reconstruction methods (e.g., pixelSplat, MVSplat, NoPoSplat, LSM) primarily utilized two-view images as input. We **adopted the same setting as these prior works to maintain consistency**. Additionally, we acknowledge that recent feed-forward multi-view reconstruction methods (e.g., VGGT) can be regarded as our **concurrent works**.
>
> **[W3] (1) Multi-task training without both R$\to$U and U$\to$R, and (2) independent training for each task.**
> Thanks for the constructive suggestion.
>
> (1) **We implement a variant model w/o R$\to$U and U$\to$R**, and show the comparison results in the table below. We can see that adding either module alone outperforms this variant model, demonstrating the effectiveness of the our R$\to$U and U$\to$R module designs.
>
> We will include the results in our revision.
>
> |Method|DepthEstimation|NovelViewSynthesis|Understanding(context-view)|Understanding(novel-view)|
> |:-:|:-:|:-:|:-:|:-:|
> |R$\to$U\|U$\to$R|AbsRel↓RMSE↓|PSNR↑SSIM↑LPIPS↓|mIoU↑mAP↑PQ↑|mIoU↑mAP↑PQ↑|
> |✘\|✘|0.096\|0.241|25.51\|0.817\|0.195|0.549\|0.236\|0.595|N/A\|N/A\|N/A|
> |✓\|✘|0.096\|0.241|25.51\|0.817\|0.195|0.589\|0.264\|0.657|0.588\|0.253\|0.646|
> |✘\|✓|0.074\|0.208|25.96\|0.822\|0.184|0.551\|0.253\|0.612|N/A\|N/A\|N/A|
> |✓\|✓|**0.074**\|**0.208**|**25.96**\|**0.822**\|**0.184**|**0.592**\|**0.282**\|**0.661**|**0.592**\|**0.271**\|**0.650**|
>
> (2) **We implement two variant models**, one with only photometric and depth continuity losses (**"Recon. Only"**), and another with only mask and text-matching losses (**"Segm. Only"**).
> For the "Recon. Only" variant, we ablate our U$\to$R module (i.e., mask-guided depth continuity loss) on it to validate its effectiveness on independent reconstruction task, where we use ground-truth masks as the depth guidance due to the absence of predicted masks. Note that we cannot add our R$\to$U module to "Segm. Only" variant without the reconstructed 3D Gaussians, thus we only implement "Segm. Only w/o R$\to$U" in the experiments. The results are shown in the table below.
>
> We can see that, for reconstruction-only task, **our U$\to$R module still significantly improves depth accuracy and novel view rendering quality**, demonstrating its effectiveness for independent reconstruction task. We can also see that, without reconstructed 3D Gaussians to help lifting multi-view masks to 3D, we cannot achieve 3D-aware segmentation on novel views. Moreover, without our R$\to$U module that can leverage 3D Gaussians to improve multi-view mask consistency, the segmentation performance would also degrade significantly. These outcomes support the insight of this paper to unify understanding and reconstruction tasks into a single framework. We will include the results in our revision.
>
> |Method|DepthEstimation|NovelViewSynthesis|Understanding(context-view)|Understanding(novel-view)|
> |:-|:-:|:-:|:-:|:-:|
> ||AbsRel↓RMSE↓|PSNR↑LPIPS↓|mIoU↑mAP↑PQ↑|mIoU↑mAP↑PQ↑|
> |Recon.Only w/o U$\to$R|0.096\|0.241|25.51\|0.195|N/A\|N/A\|N/A|N/A\|N/A\|N/A|
> |Recon.Only w/ U$\to$R|**0.073**\|**0.207**|**26.05**\|0.187|N/A\|N/A\|N/A|N/A\|N/A\|N/A|
> |Segm.Only w/o R$\to$U|N/A\|N/A|N/A\|N/A|0.561\|0.261\|0.622|N/A\|N/A\|N/A|
> |Ours full|0.074\|0.208|25.96\|**0.184**|**0.592**\|**0.282**\|**0.661**|**0.592**\|**0.271**\|**0.650**|
>
> **[W4] Narrow down the claim and research scope.**
> Thanks for the constructive suggestion.
> Indeed, this paper primarily focuses on multi-task 3D segmentation, which falls under the narrow scope of fine-grained 3D understanding tasks and does not include broader tasks such as 3D captioning and 3D question answering.
> To address this, **we will narrow down our claim by replacing all occurrences of "3D understanding" with "multi-task 3D segmentation" in our revision**.
>
> Note that our initial claim on 3D understanding is inherited from previous works (e.g., OpenGaussian[1], FMGS[2]) that also focus on reconstruction-based segmentation, which is inappropriate given the scope of 3D understanding is broader than before.
>
> [1]  OpenGaussian: Towards Point-Level 3D Gaussian based Open Vocabulary Understanding, NeurIPS'24
>
> [2] FMGS: Foundation Model Embedded 3D Gaussian Splatting for Holistic 3D Scene Understanding, IJCV'25
>
> **[Q1] What if more than two instances of the same category appear in the scene?**
> Yes. In our experiments, we observe that, if the text refers to a specific instance (e.g., "Chair nearest to the door"), no matter how many other instances of the same category appear in the scene, the segmentation result corresponding to the query with the highest matching similarity can effectively locate the instance. If the text refers to no specific instance but a category (e.g., "Chair"), the queries with topK matching similarities would find every single instance of this category, which is functionally similar to instance segmentation. We have prepared some qualitative results for verification. However, due to the updated rebuttal policy, we cannot provide here. **We promise we will include them in paper and code repository in revision.**
>
> **[Q2] Primary factors for training efficiency compared to LSM.**
> The main reason for our training efficiency compared to LSM is that we do not require high-dimensional feature rasterization to align 3D Gaussian features with 2D features. We find that such **feature-alignment paradigm requires much more training time to achieve convergence** than our alignment-free method as shown in the table below.
> |Method|AbsRel↓|PSNR↑|mIoU↑|
> |:-|:-:|:-:|:-:|
> |Ours vs. LSM(2hours train.)|0.074 vs. 0.107|25.96 vs. 19.56|0.592 vs. 0.174|
> |Ours vs. LSM(1day train.)|0.076 vs. 0.084|25.92 vs. 21.02|0.594 vs. 0.236|
> |Ours vs. LSM(3days train.)|0.076 vs. 0.075|25.89 vs. 21.88|0.595 vs. 0.271|

---

> > ### Comment · Reviewer_oS3w · 2025-08-04
> >
> > Thanks for your response! Your responses have addressed most of my concerns, and I will raise the rating to borderline accept accordingly.

---

> > > ### Author Response · Authors · 2025-08-04
> > > **Response to Comment from Reviewer#oS3w**
> > >
> > > We are grateful for your positive recommendation and the supportive feedback on our paper. Thank you very much for recognizing our efforts in addressing most of the concerns. We will incorporate the results and the corresponding clarifications into the revised version of our paper.

---

### Official Review · Reviewer_oN8Y · 2025-06-30

**Clarity:** 3
**Significance:** 2
**Originality:** 3
**Rating:** 4
**Confidence:** 4

**Summary:**

This paper presents SIU3R, an alignment-free framework that jointly performs 3D reconstruction and 3D scene understanding from two-views, unposed images. Instead of rasterizing and aligning 2D foundation-model features in 3D, SIU3R utilizes a shared set of unified queries and lifts pixel-aligned 2D predictions into a 3D Gaussian field to drives both reconstruction and understanding. Two lightweight modules (1) the Multi-View Mask Aggregation (R→U) and (2) Mask-Guided Geometry Refinement (U→R) are proposed and enable bidirectional improvements between semantics and geometry  in a single forward pass. Experiments on ScanNet demonstrate superior  performance in 3D reconstruction and scene understanding, with fast inference speed.

**Questions:**

1. In Table 2 you compare “Early Aggregate w/o train” vs. “with train” vs. “Ours”. Could you clarify exactly which weights were or weren’t updated in the “with train” variant? Was the entire network fine-tuned, or only the aggregation layer? Why reconstruction quality was worse after training.

2. In Fig 5(b) GT first row, what is the black part in the middle of the segemantic map, there is nothing in the RGB

3. You set $ \lambda_{1\ldots5}=\{1,0.5,0.05,0.05,1\}$. How did you choose these weights?

5. All visual examples (Fig 5) are from ScanNet. Could you provide some tests on in-the-wild image pairs?

**Ethical Concerns:**

["NO or VERY MINOR ethics concerns only"]

**Final Justification:**

I appreciate the authors providing a thorough response to my concerns. Most of my concerns have been addressed after the authors adding multi-view backbones and additional datasets for evaluation. I will raise my score to borderline accept.

That said, the paper still requires major revisions: particularly in motivation, backbone design, and experimental evaluation. I recommend that the authors revise it accordingly if it is accepted. However, rejecting the paper and reevaluating it through a new submission would also be acceptable

**Limitations:**

Yes

**Quality:**

3

**Strengths And Weaknesses:**

- Strength

1. Motivation: The authors highlight how prior feature‐alignment methods are both computationally intensive and prone to semantic inconsistencies, making a strong motivation for their alignment‐free framework that unifies 3D reconstruction and scene understanding.

2. Novelty: The proposed unified query decoder, R→U (mask aggregation) and U→R (geometry refinement) components are novel and interesting. These components enable instance and semantic segmentation tasks with 3D reconstruction in feedforward pass and exploit geometric and semantic cues to improve each other’s quality.

3. Multiple tasks: Across depth estimation, NVS, and a suite of 2D/3D segmentation tasks, SIU3R consistently outperforms the baselines in both reconstruction and scene understanding, show its effectiveness in ScanNet dataset.

- Weakness

1. Limited to Two Views. SIU3R uses exactly two views as input. This restriction prevents application to more general multi-view or video inputs and limits capture of global scene context.

2. Single-Dataset Training/Evaluation: All experiments are conducted on ScanNet. Given ScanNet’s noisy 3D-to-2D projections and indoor bias, it is unclear how SIU3R performs on outdoor or higher-fidelity datasets (e.g. Matterport3D, Replica, ScanNet++). Its generative reconstruction ability and segmentation robustness is not clear beyond ScanNet.

3.  If I understand correctly, the fixed set of unified queries limits SIU3R to a close-vocabulary regime, and cross-atten with text features gives it zero-shot capability. However, what happens when the number of instances in a scene is much larger than available query slots? How does the model handle more object masks than there are unified queries? Please correct me if I’ve misunderstood.

4. The comparison with other baselines are unfair, the proposed model is overfit to one dataset, finetune other baselines on ScanNet could show more fair comparison

---

> ### Author Rebuttal · Authors · 2025-07-27
>
> We thank reviewer oN8Y for recognizing our strong motivation, novelty, multi-task unification and evaluation.
> Our detailed response to the feedback is as follows.
>
> **[W1] Limited to two views.**
> Thanks for raising this concern.
>
> Building upon the insights of concurrent works on feed-forward multi-view reconstruction (e.g., VGGT), we **make minor modifications to our model to support more input views** for simultaneous understanding and 3D reconstruction.
> Specifically, compared to the model with two views, where cross-attention is only performed between the tokens of the two views, for multiple views, we perform cross-attention between the tokens of each view and the concatenated tokens of all other views. This is then followed by our Gaussian decoder and Unified Query Decoder, which predict multi-view 3D Gaussians and masks, respectively.
> We re-trained four variant models with different number of input views (i.e., 2, 4, 6, 8 views). The quantitative results, shown in the table below, demonstrate that **as the number of views increases, our model exhibits improved performance in not only reconstruction but also 3D-aware segmentation on novel views**.
> We think this improvement is primarily due to the availability of additional view information, which reduces geometry uncertainty, and enhances the quality of 3D reconstruction. This, in turn, boosts 3D scene understanding performance, thanks to our mutual-benefit mechanism (i.e., R$\rightarrow$U). We will include the quantitative results in our revision.
>
> Besides, we have also prepared some qualitative results to demonstrate our capability of handling 8 views and larger scene scales. However, due to the updated rebuttal policy, we cannot provide the results here. **We promise we will include the results in our paper and code repository in revision.**
> |No.Views|InferTime|InferVRAM|TrainVRAM|AbsRel$\downarrow$|RMSE$\downarrow$|PSNR$\uparrow$|SSIM$\uparrow$|LPIPS$\downarrow$|mIoU$\uparrow$|mAP$\uparrow$|PQ$\uparrow$|
> |:-:|:-:|:-:|:-:|:-:|:-:|:-:|:-:|:-:|:-:|:-:|:-:|
> |2|0.067s|~4G|~24G|0.074|0.208|25.96|0.822|0.184|0.592|0.271|0.650|
> |4|0.097s|~5G|~38.5G|0.072|0.205|26.28|0.822|0.179|0.611|0.283|0.687|
> |6|0.140s|~6G|~56.5G|0.071|0.203|26.45|0.830|0.175|0.617|0.287|0.693|
> |8|0.185s|~7G|~77.5G|0.071|0.201|26.65|0.838|0.170|0.620|0.289|0.697|
>
> It is also worth noting that most prior feed-forward reconstruction and understanding methods (e.g., pixelSplat, MVSplat, NoPoSplat, LSM) primarily utilized two-view images as input at the time of this paper's submission. **To maintain consistency, we adopted the same setting as these previous works in our initial submission**. Additionally, we acknowledge that recent feed-forward multi-view reconstruction methods (e.g., VGGT) can be regarded as our **concurrent works**.
>
> **[W2] Single-dataset training/evaluation.**
> Thanks for raising this insightful concern.
>
> To fully verify our performance on more datasets, we conducted two experiments, i.e. zero-shot generalization on datasets with different sensor characteristics (ScanNet++ & Replica), and train a general model with more datasets. The detailed results are shown in table below.
>
> It can be seen that SIU3R has certain zero-shot generalization capability even it is trained solely with ScanNet. However, due to the size of training data (i.e. ScanNet), its generalization performance still has rooms to improve. Despite this, our method still exhibits better generalization performance compared to other baselines under the same training settings.
>
> To further verify if SIU3R has performance gains with more training data, we train it with more datasets to get a general model. The results demonstrate that SIU3R achieved improved performance even the training data exhibits different characteristics. It demonstrates that SIU3R should achieve good performance as the dataset size further increases, as what VGGT (a concurrent work as ours, from CVPR'25) demonstrates.
>
> ||$\qquad$ScanNet|$\qquad$ScanNet++|$\qquad$Replica|
> |:-|:-:|:-:|:-:|
> |**Method**| PNSR↑AbsRel↓mIoU↑| PNSR↑AbsRel↓mIoU↑| PNSR↑AbsRel↓mIoU↑|
> |pixelSplat (train on ScanNet only)|24.93\|0.181\|N/A|17.91\|0.274\|N/A|18.12\|0.286\|N/A|
> |MVSplat (train on ScanNet only)|23.80\|0.170\|N/A|18.27\|0.261\|N/A|18.43\|0.277\|N/A|
> |NoPoSplat (train on ScanNet only)|25.91\|0.094\|N/A|19.76\|0.107\|N/A|19.88\|0.110\|N/A|
> |Ours (train on ScanNet only)|25.96\|0.074\|0.592|20.05\|0.084\|0.482|20.46\|0.087\|0.376|
> |Ours (train on ScanNet, ScanNet++, Replica)|**26.06**\|**0.073**\|**0.607**|**23.39**\|**0.074**\|**0.498**|**23.16**\|**0.079**\|**0.492**|
>
> **[W3] How does the model handle the case that the number of instances in a scene is much larger than the number of queries?**
> As desribed in Table I of our appendix, we set number of queries to 100, which denotes that the maximum number of potential objects in a scene is up to 100.
> Empirically, we observe that this setting is good enough to handle most cases in ScanNet, where **the average number of valid objects is less than 10 for two-view inputs**.
>
> However, as the scene scales increase, the number of objects actually present in the scene may exceed 100.
> In consideration of this, we **increase the number of queries from 100 to 200 to accommodate scenes with more objects**, and present the comparison results in the table below.
> Note that we conduct this experiment just for verifying that our method can accommodate more objects by increasing the number of queries. Since the average number of objects (< 10) in scenes is far fewer than 200, the redundancy in queries would incur significant query-to-mask matching ambiguity during training, which leads to slight performance degradation for segmentation tasks.
> | Method | Recon. (Depth Estimation) | Recon. (Novel View Synthesis) | Understanding (Context-View) | Understanding (Novel-View) |
> | :--- | :---: | :---: | :---: | :---: |
> |  | AbsRel $\downarrow$  RMSE $\downarrow$ | PSNR $\uparrow$ SSIM $\uparrow$ LPIPS $\downarrow$ | mIoU $\uparrow$ mAP $\uparrow$ PQ $\uparrow$ | mIoU $\uparrow$ mAP $\uparrow$ PQ $\uparrow$ |
> | 200 queries |  0.073 \| 0.206 | 25.87 \| 0.820 \| 0.191 | 0.569 \| 0.244 \| 0.658 | 0.569 \| 0.237 \| 0.647 |
> | 100 queries | 0.074 \| 0.208 | 25.96 \| 0.822 \| 0.184| 0.592 \| 0.282 \| 0.661| 0.592 \| 0.271 \| 0.650 |
>
> **[W4] Finetune other baselines on ScanNet for fair comparison.**
> As described at P7-L218, "all baseline methods are evaluated on ScanNet dataset under the same protocols as ours for fair comparison".
> To be more specific, **all of the baseline methods were re-trained on the same data as ours from scratch** following the training protocols of their official implementations.
> We think such an experimental setup can ensure that our comparison is fair.
>
> For clarity, we will highlight this in implementation details in our revision.
>
> **[Q1] Which weights were or weren't updated in "Early Aggregate w/ train"? Why reconstruction quality of "Early Aggregate w/ train" was worse than "Early Aggregate w/o train"?**
> (1) **All the weights were learnable for "Early Aggregate w/ train"** variant during training. Specifically, for this variant, we train SIU3R model from scratch and perform feature rasterization to aggregate multi-view feature information before mask decoding. By doing this, the model could learn to adjust Gaussian parameters for better multi-view aggregation.
>
> (2) As described above, all the parameters of "Early Aggregate w/ train" variant could be updated after training, which inevitably causes its performance inconsistency with "Early Aggregate w/o train".
> As for the minor performance degradation, the main reason is that **the predicted Gaussian parameters are used for not only reconstruction purpose but also feature aggregation purpose during training**, which can lead to suboptimal reconstruction quality. This also highlights the motivation of our training-free R$\rightarrow$U module.
>
> We will include the above details and discussions in our revision.
>
> **[Q2] What is the black part in the middle of the semantic map in Fig5(b)?**
> Since the segmentation masks were originally annotated on 3D meshes rather than 2D images for ScanNet dataset, the rendered ground-truth 2D masks may occasionally have some unannotated pixels (e.g., the black part in Fig5) due to the errors of 3D mesh reconstruction.
>
> **[Q3] How did you choose the loss weights?**
> We set the loss weights this way because it keeps the values of each loss at approximately the same magnitude. Empirically, we find that this configuration is optimal in most cases.
>
> **[Q4] Can you provide some tests on in-the-wild image pairs?**
> We collected some in-the-wild images using hand-held cellphones to further evaluate our generalization capability. Remarkably, our model achieves surprisingly good zero-shot performance in real-world scenes for both reconstruction and segmentation, qualitatively matching its performance on ScanNet dataset. However, image/video results are not available during the rebuttal phase due to the updated rebuttal policy. **We promise we will include the results in our paper and code repository in revision.**

---

> > ### Comment · Reviewer_oN8Y · 2025-08-03
> > **The response solve most of my concerns**
> >
> > I appreciate the authors providing a thorough response to my concerns. Most of my concerns have been addressed after the authors adding multi-view backbones and additional datasets for evaluation. I will raise my score to borderline accept.
> >
> > That said, the paper still requires major revisions: particularly in motivation, backbone design, and experimental evaluation. I recommend that the authors revise it accordingly if it is accepted. However, rejecting the paper and reevaluating it through a new submission would also be acceptable.

---

> > > ### Author Response · Authors · 2025-08-04
> > > **Responses to Comments from Reviewer#oN8Y**
> > >
> > > Thank you very much for your positive feedback and for acknowledging our efforts in addressing the majority of the concerns. We are pleased that our additional experiments have been helpful. As suggested, we will certainly integrate the results and the corresponding clarifications into the revised version of our paper. Thank you once again for your valuable comments and contributions to the review process.

---

> ### Comment · Area_Chair_gU5P · 2025-08-06
> **Clarification on Score Adjustment and Revision Guidance**
>
> Hi Reviewer oN8Y,
>
> As mentioned, you plan to improve the score, but it is currently still at borderline reject. Do you intend to increase the rating or keep the current one?
>
> In addition, it would be helpful to specify how to revise the motivation, backbone design, and experimental evaluation to make the submission stronger.
>
> Bests,
> AC

---

> > ### Comment · Reviewer_oN8Y · 2025-08-06
> > **Response to AC**
> >
> > Dear AC,
> >
> > Thanks for the reminder. For the paper revision, based on the rebuttal , the authors have built a new VGGT-based backbone to support multi-view inputs. Accordingly, the contribution and methodology sections need to be revised. Additional results on ScanNet++ and Replica should be included in the final version, and in-the-wild visualizations are also expected.
> >
> > Best,
> >
> > oN8Y

---

### Official Review · Reviewer_bcN2 · 2025-07-01

**Clarity:** 3
**Significance:** 2
**Originality:** 3
**Rating:** 4
**Confidence:** 4

**Summary:**

This paper introduces SIU3R, a novel alignment-free framework for simultaneous scene understanding and 3D reconstruction from sparse unposed images. It tackles limitations of prior methods that rely on 2D-to-3D feature alignment, which are typically slow, require dense views, and suffer from semantic degradation due to feature compression. The proposed SIU3R replaces feature alignment with a pixel-aligned 2D-to-3D lifting mechanism, combining two key modules: a Unified Query Decoder to perform multiple understanding tasks (semantic, instance, panoptic, and text-referred segmentation), and a Mutual Benefit Mechanism that fosters collaboration between reconstruction and understanding. The framework achieves state-of-the-art results across several benchmarks, including depth estimation, novel view synthesis, and 3D-aware understanding, while being efficient and generalizable.

**Questions:**

1. Baseline implementation details: Are the baselines zero-shot results or fine-tuned results? If they are zero-shot, is the comparison fair? If fine-tuned, where are the implementation details?

2. The baseline LSM is initialized from Dust3R, whereas SIU3R is initialized from MASt3R, with its unified query decoder pretrained on COCO (see Sup. L.32). This raises concerns about the fairness of the comparison.

**Ethical Concerns:**

["NO or VERY MINOR ethics concerns only"]

**Final Justification:**

I have thoroughly reviewed the comments from other reviewers and the authors' rebuttal.

The additional clarifications addressed some of my concerns (e.g., on training details and performance analysis). So **I will maintain my rating**. However, significant revisions are still needed, **particularly regarding further ablations on pretraining and the case study**. I recommend that the authors revise these areas if the paper is accepted. Alternatively, **rejecting the paper and reconsidering it through a new submission is also acceptable**.

**Limitations:**

yes

**Quality:**

3

**Strengths And Weaknesses:**

**Strengths**

1. Effectively unifies multiple 3D understanding tasks with a single set of learnable queries.

2. Clear performance gains across all benchmarks and tasks compared to strong baselines like LSM, NoPoSplat, and Mask2Former.

3. Lightweight yet effective modules (R→U and U→R) that improve results through cross-task collaboration.

**Weaknesses**

1. Generalization to other datasets: Baselines such as MVSplat, PixelSplat, and NoPoSplat also evaluate cross-dataset generalization. However, the proposed method only conducts in-domain evaluation on ScanNet.

2. Quantitative evaluation for language grounding: Only qualitative results are provided; the actual performance remains unknown.

---

> ### Author Rebuttal · Authors · 2025-07-25
>
> We thank reviewer bcN2 for acknowledging our contribution in component designs (i.e., unified queries, R$\rightarrow$U, U$\rightarrow$R modules) and performance gains (i.e., surpass strong baselines across all tasks).
> Our detailed response to the feedback is as follows.
>
> **[W1] Generalization to other datasets besides ScanNet.**
> Thanks for raising this insightful concern.
>
> To fully verify our performance on other datasets, we conducted two experiments, i.e. zero-shot generalization on datasets with different sensor characteristics (ScanNet++ & Replica), and train a general model with more datasets. The detailed results are shown in table below.
>
> It can be seen that SIU3R has certain zero-shot generalization capability even it is trained solely with ScanNet. However, due to the size of training data (i.e. ScanNet), its generalization performance still has rooms to improve. Despite this, our method still exhibits better generalization performance compared to other baselines under the same training settings.
>
> To further verify if SIU3R has performance gains with more training data, we train it with more datasets to get a general model. The results demonstrate that SIU3R achieved improved performance even the training data exhibits different characteristics. It demonstrates that SIU3R should achieve good performance as the dataset size further increases, as what VGGT (a concurrent work as ours, from CVPR'25) demonstrates.
>
> ||$\qquad$ScanNet|$\qquad$ScanNet++|$\qquad$Replica|
> |:-|:-:|:-:|:-:|
> |**Method**| PNSR↑AbsRel↓mIoU↑| PNSR↑AbsRel↓mIoU↑| PNSR↑AbsRel↓mIoU↑|
> |pixelSplat (train on ScanNet only)|24.93\|0.181\|N/A|17.91\|0.274\|N/A|18.12\|0.286\|N/A|
> |MVSplat (train on ScanNet only)|23.80\|0.170\|N/A|18.27\|0.261\|N/A|18.43\|0.277\|N/A|
> |NoPoSplat (train on ScanNet only)|25.91\|0.094\|N/A|19.76\|0.107\|N/A|19.88\|0.110\|N/A|
> |Ours (train on ScanNet only)|25.96\|0.074\|0.592|20.05\|0.084\|0.482|20.46\|0.087\|0.376|
> |Ours (train on ScanNet, ScanNet++, Replica)|**26.06**\|**0.073**\|**0.607**|**23.39**\|**0.074**\|**0.498**|**23.16**\|**0.079**\|**0.492**|
>
> **[W2] Quantitative evaluation for language grounding.**
> We note that we **have included such quantitative evaluation in Table 1 (measured by mIoU_t) of our initial manuscript**. The definition of mIoU_t can be found at P7-L241. For clarity, we will highlight in the caption that mIoU_t is the metric for language in our revision. Here we attach the results in the table below, where we can see that our SIU3R method outperforms strong baselines such as LSM and LSeg in both context-view (2D-only) and novel-view (3D-aware) language grounding tasks. Besides, the ablation study further demonstrates the effectiveness of our R$\rightarrow$U and U$\rightarrow$R modules in language grounding tasks.
> | Method | mIoU_t $\uparrow$ (context-view) | mIoU_t $\uparrow$ (target-view) |
> | :--- | :---: | :---: |
> | LSeg | 0.2127 | N/A |
> | LSM | 0.1925 | 0.1905 |
> | Ours w/o R$\rightarrow$U | 0.4572 | N/A |
> | Ours w/o U$\rightarrow$R | 0.5125 | 0.5245 |
> | Ours | **0.5273** | **0.5270** |
>
> **[Q1] Baseline implementation details.**
> As described at P7-L218, "all baseline methods are evaluated on ScanNet dataset under the same protocols as ours for fair comparison".
>
> To be more specific, for pixelSplat, MVSplat, NoPoSplat, Mask2Former, and LSM, **we re-train their models following the training protocols of their official implementations using the processed ScanNet dataset same as ours**. For reconstruction-only methods (i.e., pixelSplat, MVSplat, NoPoSplat), we only use their original rendering losses for supervision. For understanding-only method (i.e., Mask2Former), we only use its mask losses for supervision. For LSeg, since it already possesses general vision-language understanding capabilities, we directly adopt its pre-trained weights for evaluation following previous feature-alignment-based 3D understanding methods such as LSM and LangSplat [1]. In our evaluation, all baseline methods are evaluated on the same validation set as ours. For clarity, we will add these details in our revision.
>
> [1] LangSplat: 3D Language Gaussian Splatting, CVPR 2024
>
> **[Q2] Ablations on initialization and pre-training for fair comparison with LSM.**
> Thanks for the constructive suggestion. We agree that different initialization and pre-training strategies may incur different performance outcomes.
> By treating initialization and pre-training as two varying factors and combining them in different ways, **we obtained the following four variants of SIU3R to fully investigate the effects**: (1) SIU3R w/ DUSt3R initialization and w/o COCO pre-training, (2) SIU3R w/ MASt3R initialization and w/o COCO pre-training, (3) SIU3R w/ DUSt3R initialization and w/ COCO pre-training, (4) SIU3R w/ MASt3R initialization and w/ COCO pre-training (original version).
>
> The comparison is shown in the table below. We can see that the metrics for reconstruction exhibit similar results in terms of novel view synthesis, indicating that the impact of different initializations (via either Mast3R or Dust3R) on performance is not particularly significant for NVS. Furthermore, SIU3R performs consistently better than LSM in terms of NVS for either initialization.
>
> Since neither MASt3R nor DUSt3R has been pre-trained on any understanding tasks, directly using their weights trained solely on the reconstruction task leads to a significant decline in segmentation performance. Despite this, SIU3R still performs competitively as LSM even not pretrained on COCO.
>
> This aligns with the knowledge in the field of vision-based understanding. For 2D detection/segmentation methods (e.g., DETR [2], Mask2Former), it is common practice to use models pre-trained on COCO as initialization for tasks in vertical domains. Similarly, for 3D-level methods like LSM, their segmentation capabilities stem from LSeg, which is also pre-trained on datasets like ADE-20K [3], comparable in scale to COCO. By considering this, we humbly think the evaluations against LSM is fair in terms of COCO pre-training.
>
> It also demonstrates the importance of pre-training on large-scale 2D perception datasets for improving downstream 3D understanding tasks, which could potentially provide good insight for future works.
>
> | Method | Recon. (Depth Estimation) | Recon. (Novel View Synthesis) | Understanding (Context-View) | Understanding (Novel-View) |
> | :--- | :---: | :---: | :---: | :---: |
> |  | AbsRel $\downarrow$  RMSE $\downarrow$ | PSNR $\uparrow$ SSIM $\uparrow$ LPIPS $\downarrow$ | mIoU $\uparrow$ mAP $\uparrow$ PQ $\uparrow$ | mIoU $\uparrow$ mAP $\uparrow$ PQ $\uparrow$ |
> | LSM (DUSt3R, w/ LSeg) | 0.075 \| 0.219 | 21.88 \| 0.734 \| 0.304 | 0.275 \| N/A \| N/A | 0.271 \| N/A \| N/A |
> | Ours (DUSt3R, w/o COCO) | 0.092 \| 0.257 | 25.31 \| 0.802 \| 0.190 | 0.251 \| 0.150 \| 0.252 | 0.250 \| 0.160 \| 0.241 |
> | Ours (MASt3R, w/o COCO) | 0.101 \| 0.260 | 25.48 \| 0.805 \| 0.192 | 0.242 \| 0.151 \| 0.253 | 0.250 \| 0.164 \| 0.242 |
> | Ours (DUSt3R, w/ COCO) | **0.073** \| **0.206** | 25.51 \| 0.810 \| 0.198 | **0.596** \| 0.278 \| 0.657 | **0.597** \| 0.268 \| 0.645 |
> | Ours (MASt3R, w/ COCO) | 0.074 \| 0.208 | **25.96** \| **0.822** \| **0.184** | 0.592 \| **0.282** \| **0.661** | 0.592 \| **0.271** \| **0.650** |
>
> [2] End-to-End Object Detection with Transformers, ECCV 2020
>
> [3] Scene Parsing through ADE20K Dataset, CVPR 2017

---

> > ### Comment · Reviewer_bcN2 · 2025-08-03
> > **Ablation Clarity: Dataset Sizes and Pretrained Weights**
> >
> > [Q2] Ablations on initialization and pre-training for fair comparison with LSM.
> >
> > Could you provide a table comparing the pretraining datasets (e.g., number of images and segments) and the finetuning 3D datasets (e.g., number of scenes) used for each method? It seems that COCO pretraining plays a significant role in improving reconstruction and understanding performance. If available, could you also include results based on ADE20K pretraining for comparison?

---

> ### Author Response · Authors · 2025-08-05
>
> We sincerely appreciate the reviewer's feedback.
>
> As shown in the table below, we compared the statistics between different datasets used for pre-training and fine-tuning. We can see that, COCO has a larger number of images (i.e., No. Images = 123k) but relatively low segment density (i.e., No. Segments / Image = 7.3). The ADE20K dataset, has fewer images (i.e., No. Images = 22k) but a much higher segment density (i.e., No. Segments / Image = 19.6). Besides, the ScanNet dataset used for fine-tuning has 1513 scenes and 3.4 segments per image in average.
>
> | Datasets | No. Images | No. Segments | No. Segments / Image |
> |:---:|:---:|:---:|:---:|
> | COCO | ~123k | ~886k | 7.3 |
> | ADE20K | ~22k | ~435k | 19.6 |
>
> As suggested, we replaced the pre-training dataset from COCO to ADE20K and performed fine-tuning on ScanNet, with the results shown in the table below.
> We can observe that pre-training on ADE20K achieves comparable performance on both reconstruction and understanding tasks compared to pre-training on COCO, and performs better than LSM which is also based on ADE20K pre-trained model. This indicates that, in terms of pre-training, the impact of ADE20K and COCO on downstream tasks is not significantly different. This also implies that our performance advantage over LSM is primarily attributed to the design of our method and the alignment-free paradigm.
>
> |Method|Pre-training dataset|Initialization|Recon. (Depth Estimation)AbsRel↓ RMSE↓|Recon. (NovelViewSynthesis)PSNR↑ SSIM↑ LPIPS↓|Understanding(Context-View) mIoU↑ mAP↑ PQ↑|Understanding(Novel-view) mIoU↑ mAP↑ PQ↑|
> |:---:|:---:|:---:|:---:|:---:|:---:|:---:|
> |LSM|ADE20K|DUSt3R|0.075\|0.219|21.88\|0.734\|0.304|0.275\|N/A\|N/A|0.271\|N/A\|N/A|
> |Ours|N/A|DUSt3R|0.092\|0.257|25.31\|0.802\|0.190|0.251\|0.150\|0.252|0.250\|0.160\|0.241|
> |Ours|N/A|MASt3R|0.101\|0.260|25.48\|0.805\|0.192|0.242\|0.151\|0.253|0.250\|0.164\|0.242|
> |Ours|COCO|DUSt3R|0.073\|0.206|25.51\|0.810\|0.198|0.596\|0.278\|0.657|0.597\|0.268\|0.645|
> |Ours|COCO|MASt3R|0.074\|0.208|25.96\|0.822\|0.184|0.592\|0.282\|0.662|0.592\|0.271\|0.650|
> |Ours|ADE20K|DUSt3R|0.073\|0.206|25.43\|0.809\|0.199|0.575\|0.268\|0.654|0.575\|0.256\|0.642|
> |Ours|ADE20K|MASt3R|0.073\|0.206|25.77\|0.818\|0.192|0.589\|0.270\|0.644|0.588\|0.258\|0.630|

---

### Decision · Program_Chairs · 2025-09-17

**Decision:**

Accept (spotlight)

**Comment:**

This paper introduces SIU3R, the first alignment-free framework for simultaneous scene, instance understanding, and 3D reconstruction from unposed images. It addresses an important and timely problem by leveraging pixel-to-3D feature lifting. The proposed unified query design and mutual-benefit modules are effective, yielding consistent improvements across reconstruction and multi-task understanding. Reviewers initially raised concerns regarding generalization, fairness of comparisons, and limited evaluation, but the authors’ rebuttal provided additional experiments and clarifications that satisfactorily resolved these issues.   Overall, the work is technically sound, impactful, and clearly addresses a new problem. I recommend acceptance.